# Osteocalcin is necessary for the alignment of apatite crystallites, but not glucose metabolism, testosterone synthesis, or muscle mass

Takeshi Moriishi[1], Ryosuke Ozasa[2], Takuya Ishimoto[2], Takayoshi Nakano[2], Tomoka Hasegawa[3], Toshihiro Miyazaki[1], Wenguang Liu[4,5], Ryo Fukuyama[6], Yuying Wang[4,7], Hisato Komori[4], Xin Qin[4], Norio Amizuka[3], Toshihisa Komori[4]*

1 Department of Cell Biology, Nagasaki University Graduate School of Biomedical Sciences, Nagasaki, Japan, 2 Division of Materials and Manufacturing Science, Graduate School of Engineering, Osaka University, Suita, Osaka, Japan, 3 Department of Developmental Biology of Hard Tissue, Division of Oral Health Science, Graduate School of Dental Medicine, Hokkaido University, Sapporo, Japan, 4 Basic and Translational Research Center for Hard Tissue Disease, Nagasaki University Graduate School of Biomedical Sciences, Nagasaki, Japan, 5 Institute of Genetics and Cytology, Northeast Normal University, Changchun, China, 6 Laboratory of Pharmacology, Hiroshima International University, Kure, Japan, 7 College of Life and Environmental Science, Hangzhou Normal University, Hangzhou, China

* komorit@nagasaki-u.ac.jp

**Data Availability Statement:** All relevant data are within the manuscript and its Supporting Information files.

## Abstract

The strength of bone depends on bone quantity and quality. Osteocalcin (Ocn) is the most abundant noncollagenous protein in bone and is produced by osteoblasts. It has been previously claimed that Ocn inhibits bone formation and also functions as a hormone to regulate insulin secretion in the pancreas, testosterone synthesis in the testes, and muscle mass. We generated Ocn-deficient (Ocn$^{-/-}$) mice by deleting *Bglap* and *Bglap2*. Analysis of Ocn$^{-/-}$ mice revealed that Ocn is not involved in the regulation of bone quantity, glucose metabolism, testosterone synthesis, or muscle mass. The orientation degree of collagen fibrils and size of biological apatite (BAp) crystallites in the *c*-axis were normal in the Ocn$^{-/-}$ bone. However, the crystallographic orientation of the BAp *c*-axis, which is normally parallel to collagen fibrils, was severely disrupted, resulting in reduced bone strength. These results demonstrate that Ocn is required for bone quality and strength by adjusting the alignment of BAp crystallites parallel to collagen fibrils; but it does not function as a hormone.

## Author summary

The strength of bone depends on both its quantity and quality. Osteocalcin (Ocn) is the most abundant non-collagenous protein in bone, but its function remains unclear. Earlier studies by other investigators have suggested that Ocn decreases the quantity of bone by decreasing bone formation; and in addition it works as a hormone to regulate glucose metabolism, testosterone synthesis, and muscle mass in distant tissues. We have generated Ocn-deficient mice and show herein that Ocn is not required for bone formation. It is,

**Funding:** This work was supported by grants from the Japanese Ministry of Education, Culture, Sports, Science and Technology (https://www.jsps.go.jp/) to TK (Grant number: 18H05283) and to TM (18K09070). The funders had no role in study design, data collection and analysis, decision to publish, or preparation of the manuscript.

**Competing interests:** The authors have declared that no competing interests exist.

however, required for optimal bone quality and strength. Specifically, we show that in the Ocn-deficient mice collagen fibers align normally, but apatite crystallites align randomly against collagen, resulting in disorganized mineralization and reduced bone strength. Furthermore, we show that glucose metabolism, testosterone synthesis, and muscle mass are normal in the Ocn-deficient mice. We conclude that Ocn acts in bone to optimize its quality and strength, but not quantity. And, in contrast to earlier claims, it does not work as a hormone to control glucose metabolism, testosterone synthesis, and muscle mass.

## Introduction

The strength of bone is dependent on its quantity and quality. Bone quantity is determined by both bone volume and bone mineral density (BMD), which is the amount of mineral per volume unit. Several parameters, including architecture and geometry, bone turnover, cortical porosity, damage, mineralization, and the properties of minerals, collagen, and non-collagenous proteins, have been proposed as the determinants of bone quality [1]. Non-collagenous proteins play significant roles in the structural organization of bone and influence its mechanical properties [2]. Ocn is synthesized by osteoblasts, and is the most abundant non-collagenous protein in bone. The three glutamic acid residues of the protein are carboxylated. Carboxylated Ocn has high affinity for $Ca^{2+}$ and adopts an α-helical conformation by binding to $Ca^{2+}$, whereas uncarboxylated Ocn has no affinity to $Ca^{2+}$. Carboxylated Ocn has been implicated in mineralization, inhibition of hydroxyapatite growth, and chemotactic activity of osteoclast precursors [3].

One Ocn gene (*BGLAP*) has been identified in humans, while mice have a gene cluster of Ocn that consists of *Bglap*, *Bglap2*, and *Bglap3* within a 23-kb span of genomic DNA [4–6]. *Bglap* and *Bglap2* are expressed in bone, while *Bglap3* is expressed in non-osteoid tissues, including the kidneys, lungs, and male gonadal tissues [6,7]. Ocn-deficient (Ocn$^{-/-}$) mice, in which *Bglap* and *Bglap2* are deleted, show increases in trabecular and cortical bone masses and bone strength by the enhanced bone formation [8]. On the other hand, a recent study reported no difference in cortical thickness between wild-type and the same Ocn$^{-/-}$mouse line [9]. An Ocn$^{-/-}$rat was generated, and the trabecular, but not cortical bone mass was greater than that in wild-type rats and bone was stronger [10]. Fourier transform infrared microscopic (FTIR) analysis showed impaired mineral maturation in Ocn$^{-/-}$femoral cortical bone [11], whereas Raman microspectroscopic analysis revealed higher crystallinity in Ocn$^{-/-}$femoral cortical bone than that in wild-type mice [12]. Furthermore, small angle x-ray scattering (SAXS) analysis showed that the shape of mineral particles was shorter, thinner, and more disorganized in Ocn$^{-/-}$mice than in wild-type mice [13]. Hence, the functions of Ocn in bone formation, maintenance, and quality remain controversial and have yet to be clarified.

Using a single mouse model with genetic deletion of Ocn generated by Karsenty and co-workers, the authors found that uncarboxylated Ocn functions as a hormone that regulates glucose metabolism by enhancing β-cell proliferation, insulin secretion, and insulin sensitivity [14]. In spite of many clinical studies prompted by these findings, the relationship between uncarboxylated Ocn and glucose metabolism remains controversial [15–24]. Karsenty and co-workers have further shown that male Ocn$^{-/-}$mice have small testes and impaired fertility due to severely reduced testosterone synthesis, and that uncarboxylated Ocn regulates testosterone synthesis by binding to its receptor Gprc6a on Leydig cells in the testes [25]. In three different global *Gprc6a$^{-/-}$*mouse models, glucose metabolism, fertility, and BMD are inconsistent [26–30]. Furthermore, it remains controversial whether Gprc6a is involved in insulin secretion and

is a receptor for Ocn in Leydig cells in testis and β cells in pancreas [25,29,31–35]. Karsenty and co-workers have also shown that $Ocn^{-/-}$ mice have reduced muscle mass and uncarboxylated Ocn promotes the uptake and catabolism of glucose and fatty acids as well as protein synthesis in myofibers [36].

Because only one line of $Ocn^{-/-}$ mice was analyzed in previous studies, we generated $Ocn^{-/-}$ mice in which *Bglap* and *Bglap2* are deleted to clarify the controversies surrounding the bone phenotypes and extraskeletal functions of Ocn.

## Results

### Trabecular and cortical bone masses are similar in wild-type and $Ocn^{-/-}$ mice

$Ocn^{-/-}$ mice were generated by replacing genomic DNA encompassing *Bglap* and *Bglap2* with the neo gene (Fig 1A). This replacement was confirmed by Southern blot and PCR analyses (Fig 1B and 1C). Real-time RT-PCR analysis showed that the expression of *Bglap* and *Bglap2* was absent in the osteoblast fraction of bone from $Ocn^{-/-}$ mice (Fig 1D). Serum osteocalcin was also absent in $Ocn^{-/-}$ mice (Fig 1E). μ-CT analyses were performed on male mice at 14 weeks, 6 months, and 9 months of age (Fig 1F–1I) and female mice at 6 and 9 months of age (S1 Fig). Bone volume, trabecular thickness, and trabecular number in the trabecular bone of femurs were similar between wild-type and $Ocn^{-/-}$ mice in both sexes (Fig 1F and 1H; S1A and S1C Fig). In the cortical bone of femurs, cortical thickness was also similar between wild-type and $Ocn^{-/-}$ mice in both sexes (Fig 1G and 1I; S1B and S1D Fig), while periosteal and endosteal perimeters were larger in male $Ocn^{-/-}$ mice than in male wild-type mice at 6 months of age, but not at 14 weeks or 9 months of age (Fig 1I). These differences were not observed in female mice (S1D Fig).a

### Bone formation and resorption are similar in wild-type and $Ocn^{-/-}$ mice

Bone histomorphometric analysis was performed on male mice at 6 months of age. The parameters for osteoblasts, osteoclasts, osteocytes, and bone formation in the trabecular bone of femurs were similar between wild-type and $Ocn^{-/-}$ mice (Fig 2A). In cortical bone, the mineral apposition rate, mineralizing surface, and bone formation rate in the periosteum were similar between wild-type and $Ocn^{-/-}$ mice (Fig 2B), while the mineral apposition rate and bone formation rate in the endosteum were similar, whereas the mineralizing surface was smaller in $Ocn^{-/-}$ mice than in wild-type mice (Fig 2C).a

Serum markers for bone formation (N-terminal propeptide of type I procollagen: P1NP) and bone resorption (tartrate-resistant acid phosphatase 5b: TRAP5b and C-terminal telopeptide crosslink of type I collagen: CTX1) were similar between wild-type and $Ocn^{-/-}$ mice at 11 weeks, 6 months, and 9 months of age (Fig 3A). The expression of osteoblast and osteoclast marker genes, including *Col1a1*, *Spp1*, *Alpl*, *Runx2*, *Sp7*, and *Ctsk*, was similar in wild-type and $Ocn^{-/-}$ mice at 6 months of age (Fig 3B). The expression of *Bglap* and *Bglap2* was not detected in $Ocn^{-/-}$ mice (Fig 3B).a

To evaluate the function of Ocn in an estrogen-depleted state, ovariectomy was performed at 5 weeks of age and mice were analyzed at 11 weeks of age because bone loss after ovariectomy is not apparent in adult B57BL/6 mice due to a low cancellous bone mass [37]. Bone volume in trabecular bone was significantly reduced in wild-type mice and marginally reduced in $Ocn^{-/-}$ mice, and the trabecular number was significantly lower in wild-type and $Ocn^{-/-}$ mice after ovariectomy than in the respective sham-operated mice (S2A and S2C Fig). Ovariectomy did not affect cortical bone in wild-type or $Ocn^{-/-}$ mice (S2B and S2D Fig), because B57BL/6

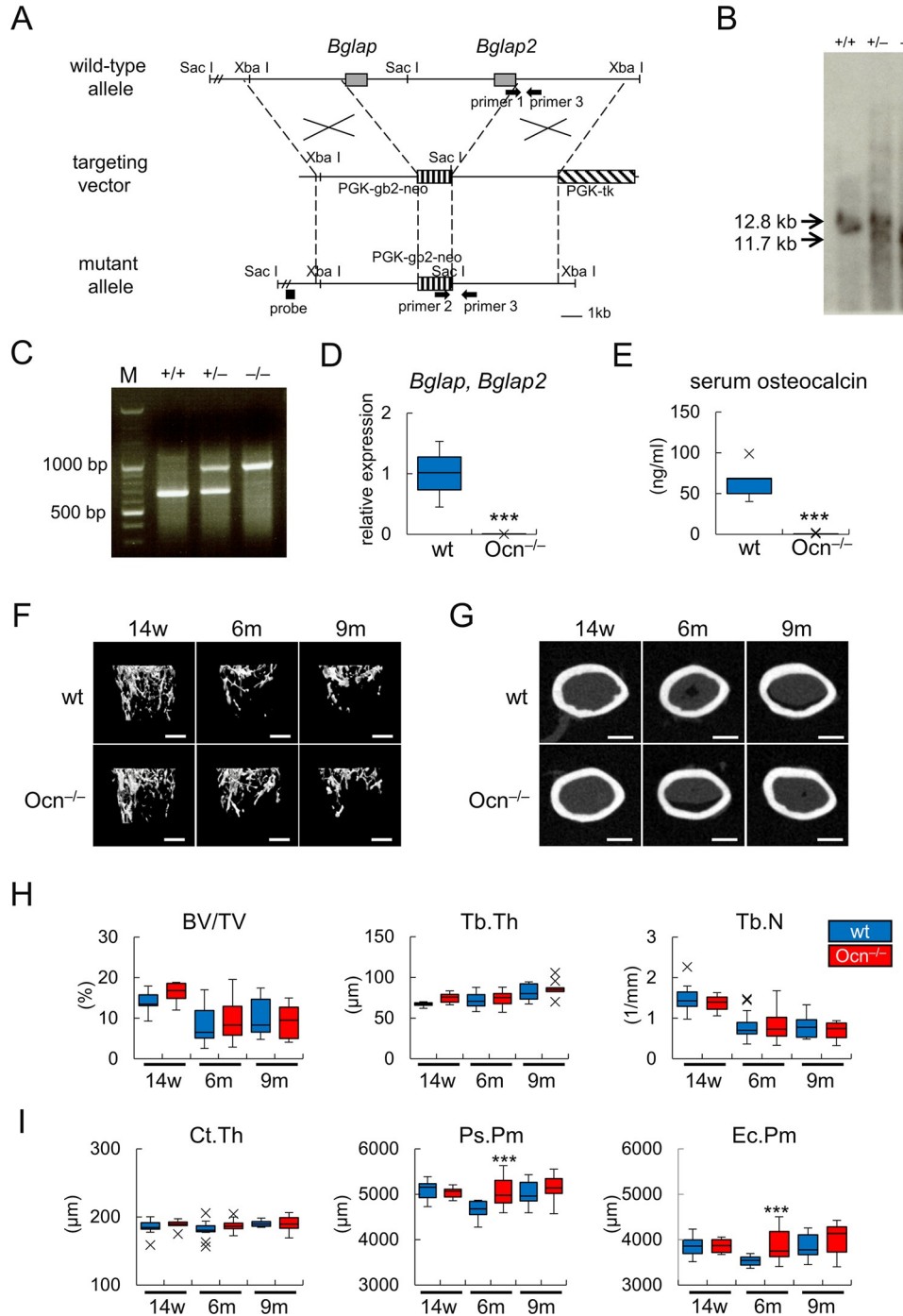

**Fig 1. Generation of Ocn$^{-/-}$ mice and a μ-CT analysis of femurs.** (A) Schematic presentation of *Bglap* and *Bglap2* gene loci, targeting vectors, and mutated alleles. (B) Southern blot analysis. DNA from tails was digested with SacI and hybridized with the probe shown in A. Bands are indicated corresponding to wild-type (12.8 kb) and mutated (11.7 kb) alleles. (C) Genotyping by PCR. PCR was performed using primers 1, 2, and 3 shown in A. Primers 1 and 3 amplify 706 bp DNA of the wild-type allele, and primers 2 and 3 amplify 980 bp DNA of the mutated allele. (D) Real-time RT-PCR analysis of *Bglap* and *Bglap2* expression using RNA from the osteoblast fractions of femurs in wild-type (wt) and Ocn$^{-/-}$ mice at 9 months of age. Primers were designed to detect the expression of both *Bglap* and *Bglap2*. wt: n = 3, Ocn$^{-/-}$: n = 7. E, The levels of serum Ocn in female mice at 11 weeks of age. wt: n = 5, Ocn$^{-/-}$: n = 7. $^{***}$P<0.001. (F-I) μ-CT analyses of femurs in male wild-type and Ocn$^{-/-}$ mice at 14 weeks, 6 months, and 9 months of age. μ-CT images of femoral distal metaphyses (F) and mid-diaphyses (G) are shown. Bars = 500 μm. H, Trabecular bone parameters, including the trabecular bone volume (BV/TV), trabecular thickness (Tb.Th), and trabecular number (Tb.N). I,

Cortical bone parameters including cortical thickness (Ct. Th), the periosteal perimeter (Ps.Pm), and endocortical perimeter (Ec.Pm). wt (n = 12), Ocn$^{-/-}$(n = 6) at 14w; wt (n = 14), Ocn$^{-/-}$ (n = 24) at 6m; wt (n = 7), Ocn$^{-/-}$ (n = 9) at 9m. *vs. wild-type mice. ***P<0.001. X symbols in box plots show outliers.

mouse line is relatively inseneitive to ovariectomy [38]. This is in contrast to previous findings, which showed more severe bone loss in Ocn$^{-/-}$ mice than in wild-type mice after ovariectomy [8]; therefore, estrogen deficiency exerted similar effects in wild-type and Ocn$^{-/-}$ bone.

## Ocn is localized in intrafibrillar and interfibrillar collagen regions and mineralization proceeds normally in Ocn$^{-/-}$ mice

Trabecular and cortical bone were similar between wild-type and Ocn$^{-/-}$ femurs in H-E-stained sections (S3 Fig). Furthermore, similar numbers of TRAP-positive cells were observed in wild-type and Ocn$^{-/-}$ femurs (S4 Fig). Transmission electron microscopy (TEM) analysis showed that collagen fibers generally ran parallel to the longitudinal direction of bone in wild-type and Ocn$^{-/-}$ mice (Fig 4A–4F). The binding of the Ocn antibody was detected in the intrafibrillar and interfibrillar collagen regions of wild-type bone, but not Ocn$^{-/-}$ bone (Fig 4A–4F). TEM analysis using non-decalcified sections revealed the presence of mineralized nodules, which are globular assemblies of numerous needle-shaped mineral crystals, in the osteoids of wild-type and Ocn$^{-/-}$ mice, indicating that mineralization proceeded similarly in wild-type and Ocn$^{-/-}$ mice (Fig 4G–4J).a

## A Raman microspectroscopic analysis showed no significant differences in mineralization, crystallinity, carbonate to phosphate ratios, collagen maturity, or the remodeling index between wild-type and Ocn$^{-/-}$ mice

The chemical aspects of bone collagen and BAp were analyzed in the posterior part of the femurs at mid-diaphysis by Raman microspectroscopy (Fig 5A and 5B). The mineral to matrix ratio, which indicates the amount of mineralization; crystallinity, which indicates the degree of mineral crystallinity; carbonate to phosphate ratio, which varies with bone architecture, age, and mineral crystallinity; collagen maturity, which assesses non-reducible to reducible collagen crosslink ratios; and the remodeling index, which indicates bone remodeling [39], were similar between wild-type and Ocn$^{-/-}$ mice (Table 1, Fig 5C).a

## The alignment of collagen fibers was parallel to the longitudinal axis of bone and the size of BAp crystallites was normal in Ocn$^{-/-}$ mice, whereas the crystallographic *c*-axis orientation of BAp was severely disrupted, leading to reduced Young's modulus

BMD was similar between female wild-type and Ocn$^{-/-}$ mice throughout the length of the femur, except at position 9 at 9 months of age (Fig 6A and 6C). To characterize the bone microstructure, the orientation of collagen was examined using birefringence measurements. The orientation of collagen was almost parallel to the longitudinal axis of bone in the diaphysis and gradually became disturbed from the metaphysis to epiphysis at similar levels in wild-type and Ocn$^{-/-}$ femurs (Fig 6B and 6D). The *c*-axis orientation degree of BAp that grows parallel to collagen in normal mineralization was analyzed using a microbeam X-ray diffraction (μXRD) system, which provides information on the atomic arrangement of crystalline apatite [40]. The intensity ratio of (002)/(310), an index representing the preferential orientation degree of the BAp *c*-axis, parallel to the bone longitudinal axis, was markedly lower in Ocn$^{-/-}$ femurs than in wild-type femurs (Fig 6E). Regression analysis showed that the BAp *c*-axis orientation degree

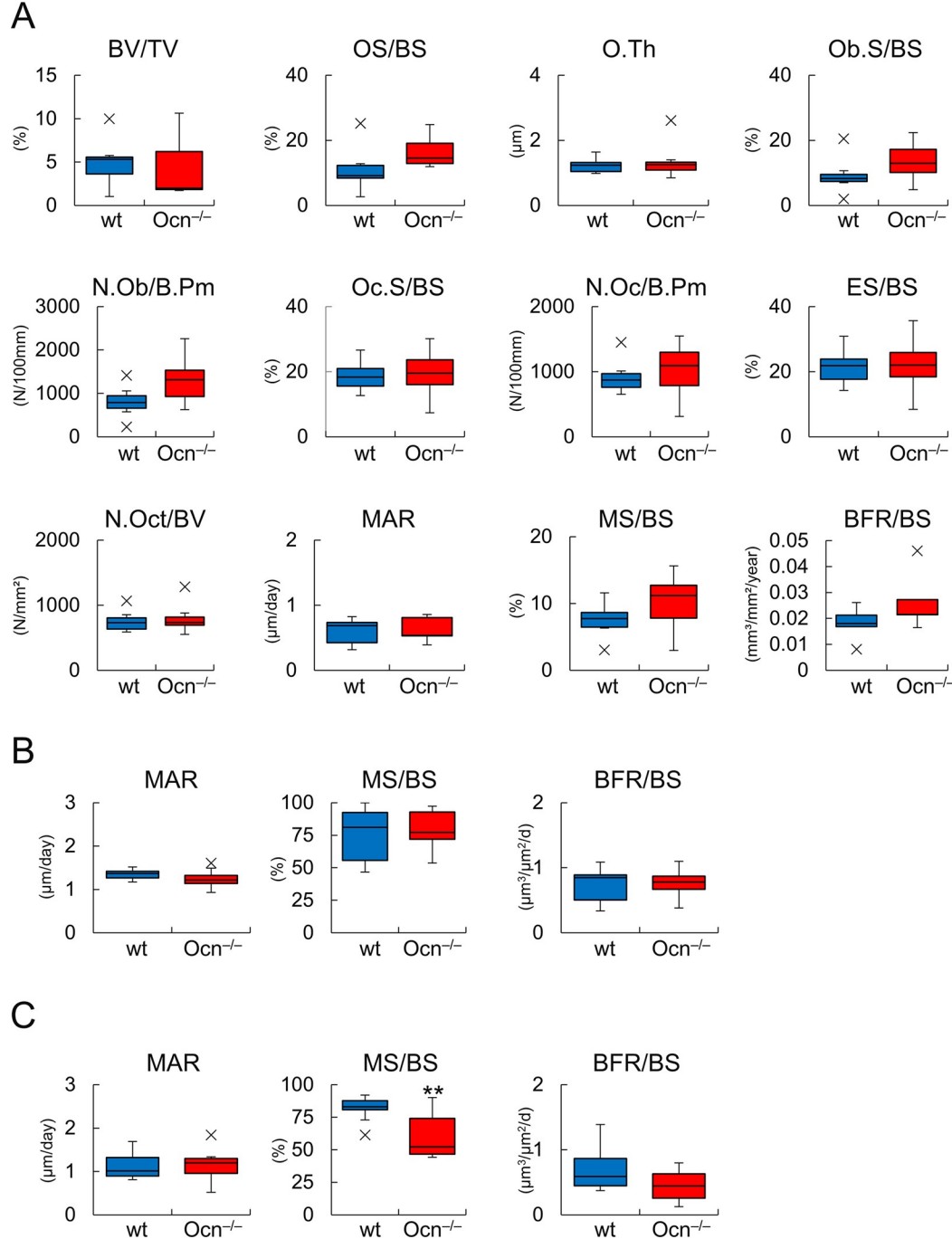

**Fig 2. Bone histomorphometric analysis.** (A) Bone histomorphometric analysis of trabecular bone in femurs at 6 months of age. The trabecular bone volume (bone volume/tissue volume, BV/TV), osteoid surface (OS/BS), osteoid thickness (O. Th), osteoblast surface (Ob.S/BS), number of osteoblasts (N.Ob/B.Pm), osteoclast surface (Oc.S/BS), number of osteoclasts (N.Oc/B.Pm), eroded surface (ES/BS), number of osteocytes (N.Oct/BV), mineral apposition rate (MAR), mineralizing surface (MS/BS), and bone formation rate (BFR/BS) were compared between male wild-type (blue column, n = 7) and Ocn$^{-/-}$ (red column, n = 7) mice. B.Pm, bone perimeter; BS, bone surface. (B and C) Dynamic histomorphometric analysis of the periosteum (B) and endosteum (C) in the mid-diaphyses of femoral cortical bones in male wild-type (n = 10) and Ocn$^{-/-}$(n = 12) mice at 6 months of age. * vs. wild-type mice. **P<0. 01. X symbols in box plots show outliers.

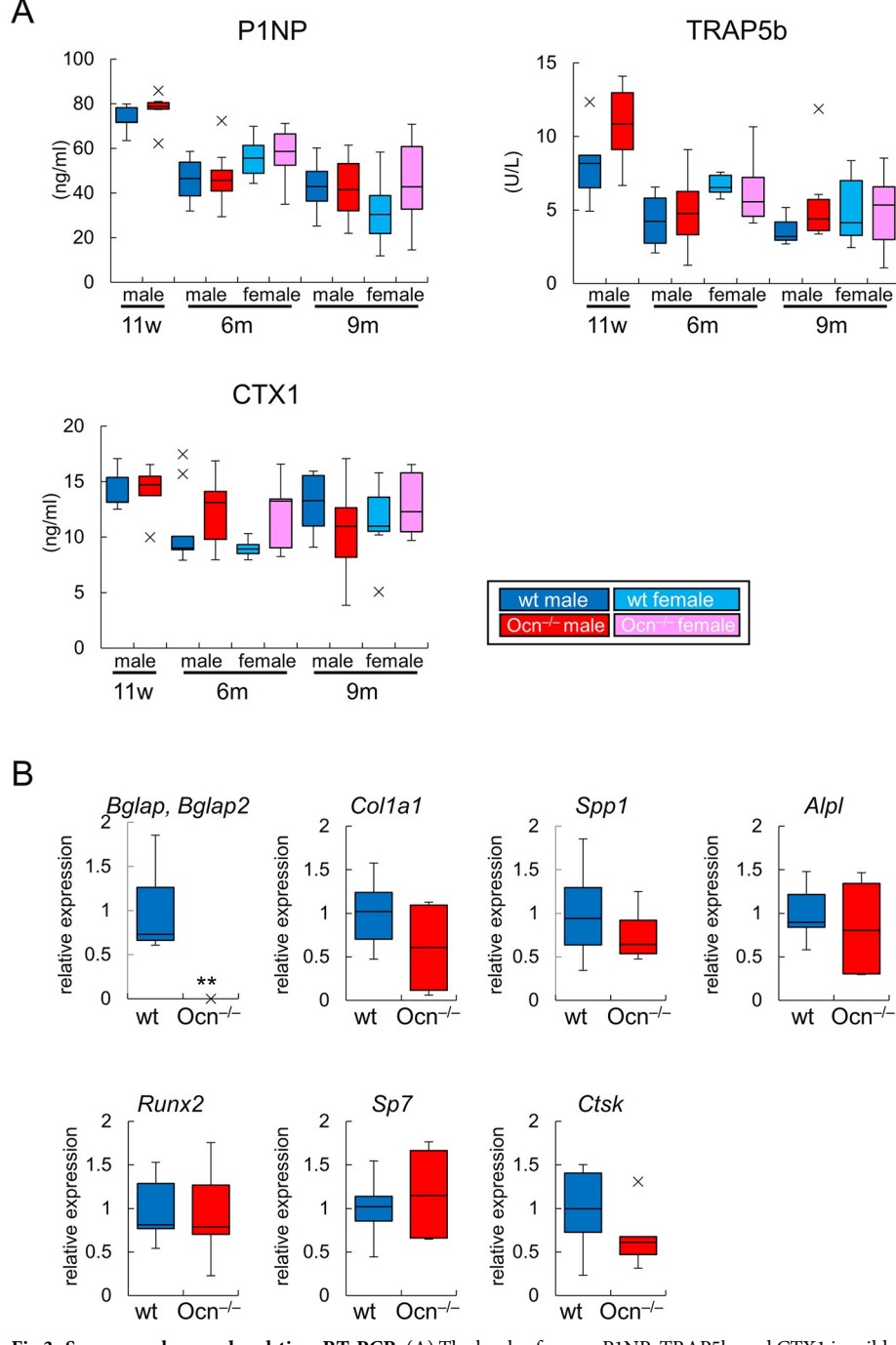

**Fig 3. Serum markers and real-time RT-PCR.** (A) The levels of serum P1NP, TRAP5b, and CTX1 in wild-type (wt) and Ocn$^{-/-}$ mice at 11 weeks, 6 months, and 9 months of age. Male 11 weeks, wt (n = 5) and Ocn$^{-/-}$(n = 6); male 6 months, wt (n = 10) and Ocn$^{-/-}$(n = 10); female 6 months, wt (n = 6) and Ocn$^{-/-}$(n = 8); male 9 months, wt (n = 5) and Ocn$^{-/-}$(n = 6); female 9 months, wt (n = 4) and Ocn$^{-/-}$(n = 7). (B) Real-time RT-PCR analysis. RNA was extracted from osteoblast fractions from femurs in male wild-type and Ocn$^{-/-}$ mice at 6 months of age. The values in wild-type mice were defined as 1, and relative levels are shown. wt: n = 7, Ocn$^{-/-}$: n = 6. **: P<0.01. X symbols in box plots show outliers.

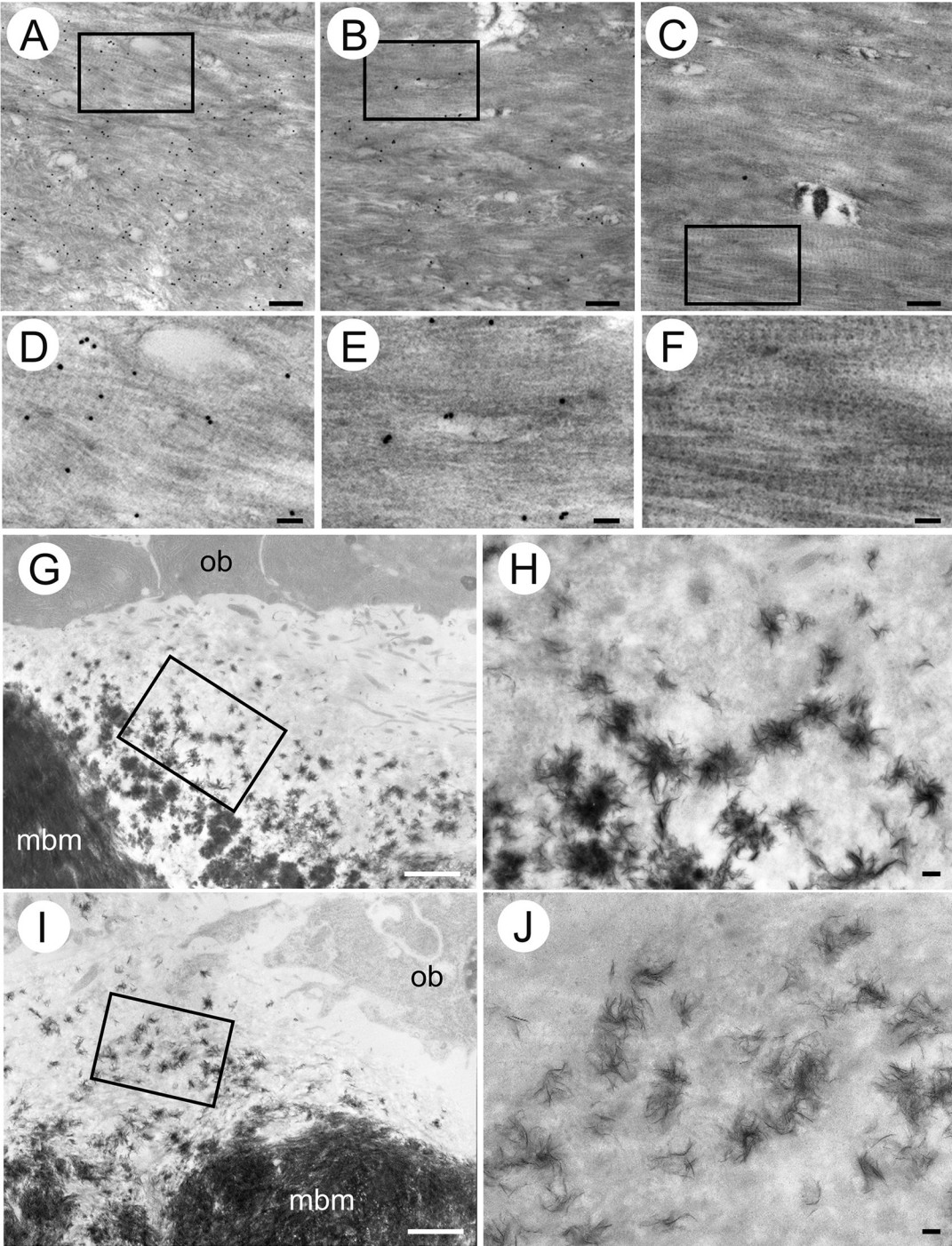

**Fig 4. TEM analyses of cortical bone.** (A-F) Immunoelectron microscopic analysis. Ultrathin sections from a wild-type femur at 10 days of age (A, D) and humerus at 4 months of age (B, E), and Ocn$^{-/-}$humerus at 4 months of age (C, F) were immunolabeled with an anti-osteocalcin antibody and analyzed by TEM. Ocn molecules immunolabeled with 20-nm gold particles were detected in wild-type (A, B, D, E), but not Ocn$^{-/-}$(C, F) bone. Boxed regions in A, B, and C are magnified in D, E, and F, respectively. The region about 8 μm from osteoid is shown in A, and those about 2 μm from osteoid are shown in B and C. (G-J) TEM analysis of non-decalcified sections. The osteoid regions of the metaphyses of tibiae in wild-type (G, H) and Ocn$^{-/-}$(I, J) mice at 4 weeks of age. The boxed regions in G and I are magnified in H and J, respectively. ob: osteoblast, mbm: mineralized bone matrix. H and J show mineralized nodules in osteoids. Bars: 500 nm (A-C, H, J), 100 nm (D-F), and 5 μm (G, I).

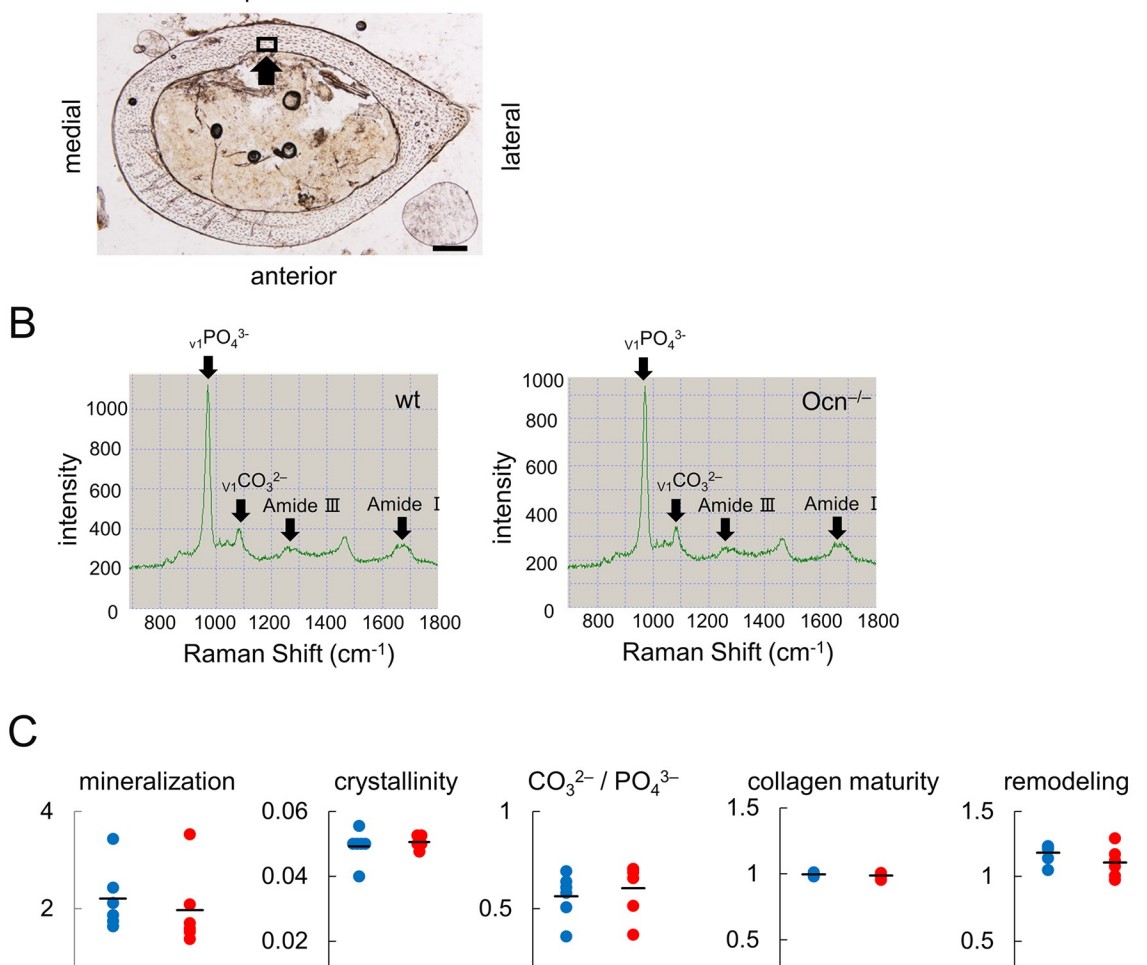

**Fig 5. Raman microspectroscopic analysis of femurs in male wild-type and Ocn$^{-/-}$ mice at 14 weeks of age.** (A) A bright field image of the section at mid-diaphysis. The analyzed region is boxed (arrow). (B) Raman spectrum showing the assignments for PO$_4^{3-}$ (959cm$^{-1}$), CO$_3^{2-}$ (1070 cm$^{-1}$), Amide III (1243–1320 cm$^{-1}$), and Amide I (1616–1720 cm$^{-1}$). (C) Parameters including mineralization, crystallinity, CO$_3^{2-}$/PO$_4^{3-}$, collagen maturity, and remodeling were calculated as shown in Table 1. n = 6.

largely depended on the orientation degree of collagen in wild-type femurs, while this dependency (slope) was weaker in Ocn$^{-/-}$ femurs, demonstrating that the epitaxial relationship between collagen and BAp was significantly disrupted in Ocn$^{-/-}$ mice (Fig 6F). However, the

**Table 1. Definition of the parameters for Raman microspectroscopic analysis.**

| mineral to matrix ratio | : | $_{v1}$PO$_4^{3-}$ / Amide |
|---|---|---|
| crystallinity | : | bandwidth (full width at half maximum) of the $_{v1}$PO$_4^{3-}$ (958 cm$^{-1}$) peak |
| carbonate to phosphate ratio | : | carbonate (CO$_3^{2-}$) / PO$_4^{3-}$ |
| collagen maturity | : | collagen crosslink ratios (1660/1690 cm$^{-1}$ ratios) |
| remodeling index | : | carbonate ($_{v1}$CO$_3^{2-}$) / Amide |

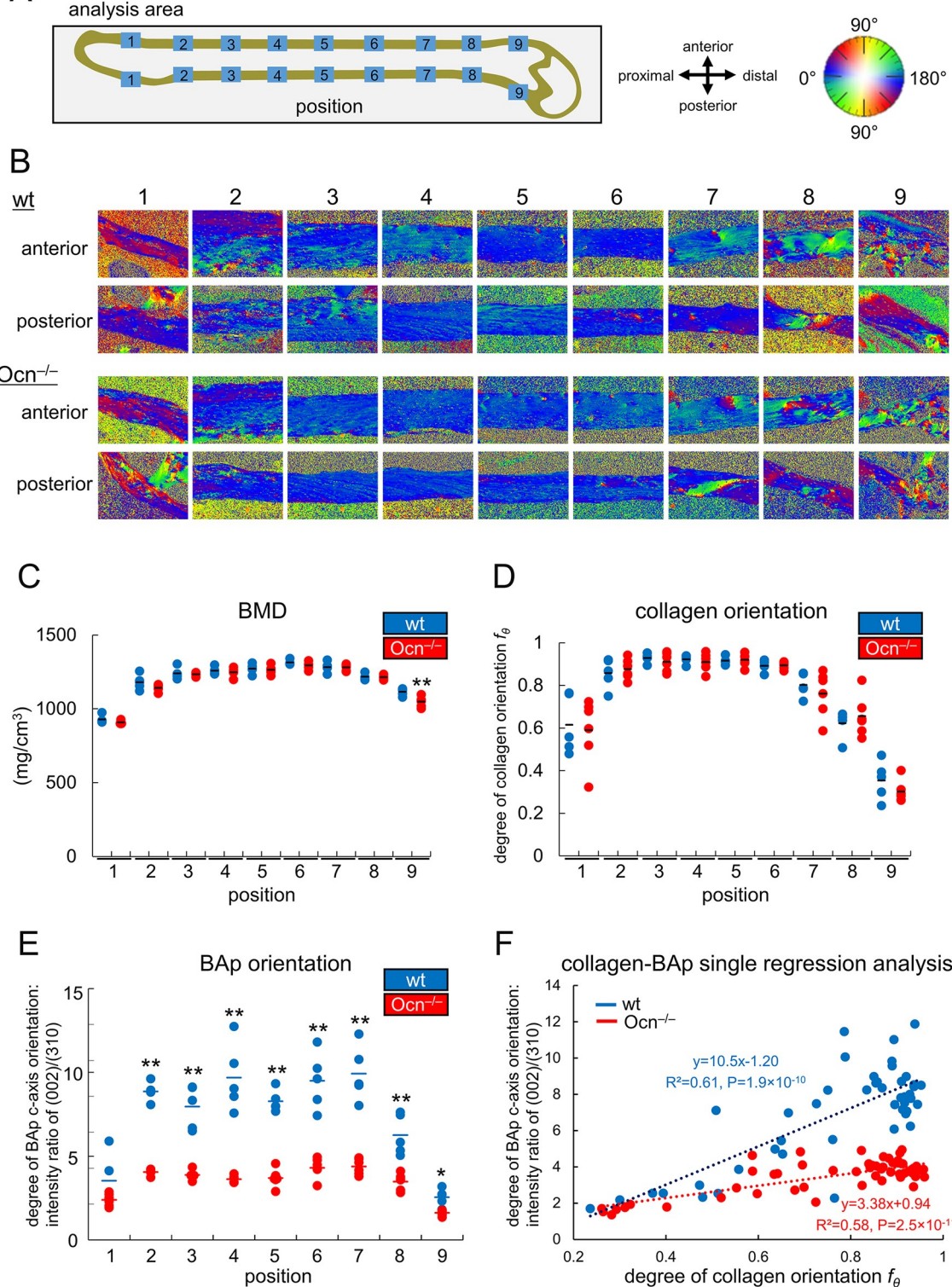

**Fig 6. BMD and orientations of collagen fibers and the BAp *c*-axis in the femoral cortical bone of female wild-type and Ocn^−/^**
**^−^mice at 9 months of age.** (A) Schematic presentation of analyzed positions and the correlation of angles and colors used for
collagen orientation in B. (B) The orientation of collagen fibers shown in color. (C) BMD. (D) Collagen orientation degree. If the
orientation of collagen fibers is completely parallel to the longitudinal direction of bone, the degree is one. (E) BAp *c*-axis orientation
degree. The preferential alignment of the BAp *c*-axis along the bone longitudinal direction was the intensity ratio of the (002)

diffraction peak to the (310) peak. Higher values indicate a more preferential alignment to the longitudinal direction. *: $P<0.05$, **: $P<0.01$. (F) Single regression analysis of the orientation of collagen fibers and the BAp $c$-axis. wt: blue dots, Ocn$^{-/-}$: red dots. wt: n = 5, Ocn$^{-/-}$: n = 6 in C-F.

size of BAp crystallites in the $c$-axis evaluated along the longitudinal and tangential axes of bone at mid-diaphysis was similar between wild-type and Ocn$^{-/-}$ femurs (Fig 7A). Similar results were obtained in male mice at 14 weeks of age for BMD and the orientations of collagen and BAp as well as the correlation between the orientations of collagen and the BAp $c$-axis (S5A–S5E Fig).aa

To evaluate bone mechanical functions in the major loading direction, nanoindentation testing was performed along the bone longitudinal axis at the mid-diaphysis (position 5) in 9-month-old mouse femurs. Young's modulus was significantly lower in Ocn$^{-/-}$ femurs than in wild-type femurs (Fig 7B). Single and multiple regression analyses were performed using Young's modulus as a dependent variable and BMD, the collagen orientation degree, and apatite $c$-axis orientation degree as explanatory variables, all of which were taken at position 5. Young's modulus positively correlated with the BAp $c$-axis orientation degree, but not with BMD or the orientation of collagen in the single regression analysis (Fig 7C). A multiple regression analysis showed that Young's modulus was strongly and solely influenced by the orientation of BAp, but not by that of collagen or BMD (Table 2). The reduction in Young's modulus in Ocn$^{-/-}$ mice and the importance of the orientation of BAp to Young's modulus was also revealed in male mice at 14 weeks of age (S5F and S5G Fig, S1 Table). However, three-point bending test did not show any difference between wild-type and Ocn$^{-/-}$ mice in maximum load, displacement, stiffness, and energy to failure at 6 months of age, indicating that the three-point bending test is less sensitive than the nanoindentation testing for the evaluation of bone mechanical functions in the major loading direction (S6 Fig).

## Glucose metabolism is normal in Ocn$^{-/-}$ mice

Body weights were similar between wild-type and Ocn$^{-/-}$ mice in both sexes from 11 weeks to 18 months of age, except for female Ocn$^{-/-}$ mice at 9 months of age (Fig 8A). Blood glucose and HbA1c levels were also similar between wild-type and Ocn$^{-/-}$ mice in both sexes at all ages (Fig 8B and 8C). Furthermore, the amounts of subcutaneous and visceral adipose tissues in Ocn$^{-/-}$ mice were similar to those in wild-type mice in 14 week-old male and 9-month-old female mice (Fig 8D–8H). Glucose tolerance tests (GTTs) were performed by injecting glucose intraperitoneally into mice fed a normal or high-fat diet in both sexes from 14 weeks to 18 months of age. In all ages, both sexes, and both diets, serum glucose and insulin levels were similar between wild-type and Ocn$^{-/-}$ mice (Fig 9A–9E, S7 Fig). Further, insulin tolerance tests (ITTs) were performed in male mice fed a normal or high-fat diet at 4 and 8 months of age. Serum glucose levels were also similar between wild-type and Ocn$^{-/-}$ mice (Fig 9F and 9G). These results indicate that Ocn is not physiologically involved in glucose metabolism.aa

## Serum levels of carboxylated Ocn but not uncarboxylated Ocn are associated with an increase of bone formation and improvement of glucose metabolism by exercise

Since Ocn is one of the markers for bone formation and exercise affects glucose metabolism and bone formation, we examined the relationships among exercise, glucose metabolism, and bone formation with a treadmill. In wild-type mice of the C57BL/6 strain with a normal diet, body weight, but not HbA1c, was lower in the exercise group than in the control group (Fig

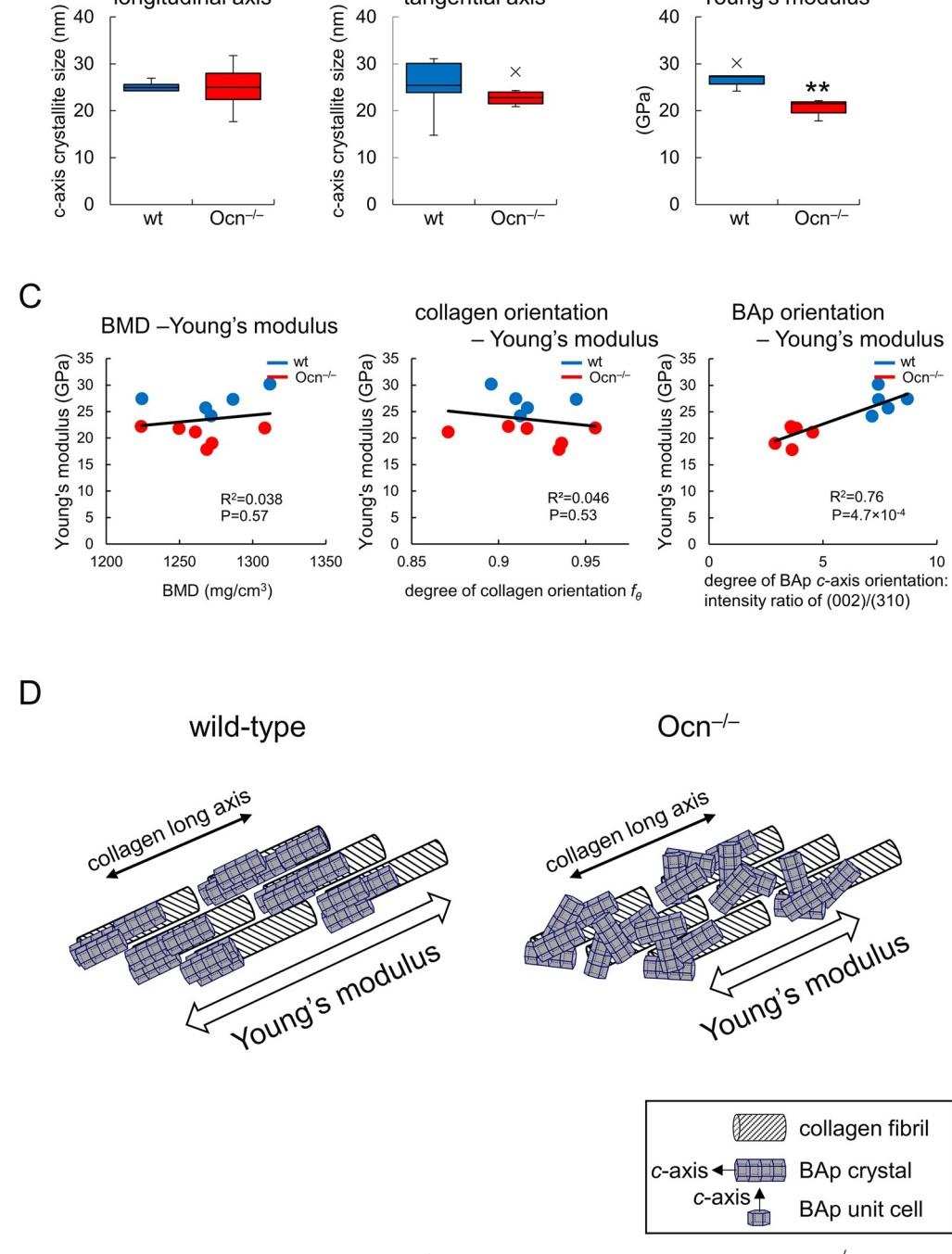

**Fig 7. Crystallite size in the BAp *c*-axis and Young's modulus in femurs of female wild-type and Ocn$^{-/-}$ mice at 9 months of age.** (A) The crystallite size in the BAp *c*-axis in the longitudinal and vertical directions at position 5. (B) Young's modulus along the bone longitudinal axis at position 5 in Fig 6A as measured by nanoindentation. $**$: P<0.01. X symbols in box plots show outliers. (C) Single regression analysis of Young's modulus to BMD and each degree of the preferential alignment of collagen fibers and the BAp *c*-axis in the bone longitudinal direction at position 5. wt: blue dots, Ocn$^{-/-}$: red dots. wt: n = 5, Ocn$^{-/-}$: n = 6. (D) Schematic presentation of orientations of collagen fibrils and BAp *c*-axis. In wild-type mice, the orientations of collagen fibrils and BAp *c*-axis are parallel to the bone longitudinal axis. In Ocn$^{-/-}$ mice, the orientation of collagen fibrils is parallel to the bone longitudinal axis, whereas that of BAp *c*-axis is severely disrupted from the longitudinal axis. The disruption of the BAp *c*-axis orientation reduces Young's modulus in the longitudinal axis.

**Table 2. Multiple regression analysis (Young's modulus-BAp, collagen, BMD).**

| BAp orientation | | collagen orientation | | BMD | |
|---|---|---|---|---|---|
| β | P-value | β | P-value | β | P-value |
| 0.84 | $2.1 \times 10^{-3}$ | −0.12 | 0.55 | 0.19 | 0.33 |

β: standard partial regression coefficient

10A and 10B). Serum levels of carboxylated Ocn, uncarboxylated Ocn, and P1NP were higher in the exercise group than in the control group. CTX-1 was not affected by exercise (Fig 10C). μ-CT analysis of trabecular bone showed that bone volume, trabecular number, and BMD increased after exercise (Fig 10D). In a mouse model (KK/TaJcl) of diabetes mellitus with a high-fat diet, exercise decreased body weight and HbA1c, increased carboxylated Ocn and P1NP, and had no effects on uncarboxylated Ocn or CTX-1 (Fig 10E–10G). μ-CT analysis of trabecular bone showed that the trabecular number and BMD increased after exercise (Fig 10H). Cortical bone was unaffected by exercise in wild-type and KK/TaJcl mice (S8 Fig). These results indicate that exercise improves glucose metabolism and increases bone formation; similar changes were observed in the serum levels of carboxylated Ocn and P1NP after exercise, and uncarboxylated Ocn did not play a role in the improvements in glucose metabolism caused by exercise.a

## Testosterone synthesis, spermatogenesis, and muscle mass are normal in Ocn$^{-/-}$ mice

Male Ocn$^{-/-}$ mice were fertile and had testes of a similar size and weight to those of wild-type mice (Fig 11A and 11B). Furthermore, serum testosterone levels were similar between wild-type and Ocn$^{-/-}$ mice at 11 weeks, 6 months, and 18 months of age (Fig 11C). Histological analysis showed that spermatogenesis proceeded normally in the testes of Ocn$^{-/-}$ mice and the number of spermatozoa was similar between wild-type and Ocn$^{-/-}$ mice (Fig 11D and 11E, S9A–S9J Fig). Moreover, the frequencies of the spermatozoa with acrosomal defects were similar between them (Fig 11F and 11G). We examined germ cell apoptosis, which is inhibited by testosterone [41,42], in the seminiferous tubules by TUNEL staining. The number of TUNEL-positive germ cells was similar between wild-type and Ocn$^{-/-}$ mice (Fig 11H and 11I). We also examined the expression of the genes, including *Star*, *Cyp11a1*, *Cyp17a1*, and *Hsd3b2*, which encode the enzymes that are necessary for testosterone biosynthesis. The expression levels of these genes in Ocn$^{-/-}$ testes were similar to those in wild-type testes (Fig 11J).a

The weights of muscles, including the quadriceps, gastrocnemius, soleus, and extensor digitorum longus, and the average area of muscle fibers in histological sections were similar between wild-type and Ocn$^{-/-}$ mice (Fig 12, S9K–S9N Fig). These results indicate that Ocn is not physiologically involved in testosterone synthesis, spermatogenesis, and the maintenance of muscle mass.a

## Discussion

In the present study, global deletion of Ocn using gene targeting in embryonic stem cells revealed that an Ocn deficiency did not affect bone mass, BMD, bone formation, or bone resorption. Furthermore, bone loss after ovariectomy was similar in wild-type and Ocn$^{-/-}$ mice. These results indicate that Ocn is not involved in the regulation of bone quantity; however, its effects on bone quality remain controversial in the literature [11–13]. The *c*-axis orientation of BAp along the femur longitudinal axis was severely disrupted in our Ocn$^{-/-}$ mice,

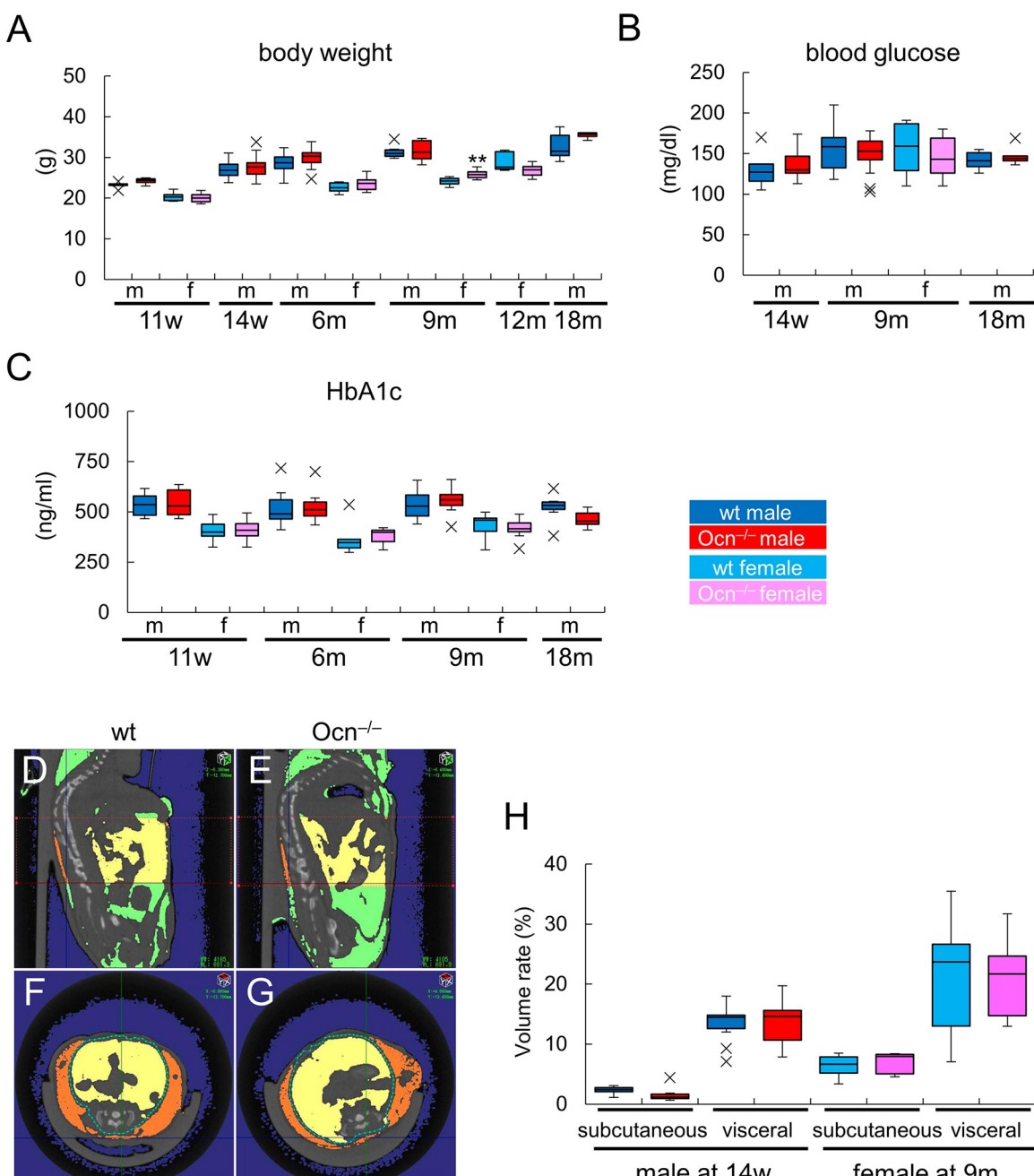

**Fig 8. Body weight, blood glucose, HbA1c, and adipose tissues in Ocn$^{-/-}$ mice.** (A)Body weights of male (m) and female (f) wild-type and Ocn$^{-/-}$ mice. Male (wt: n = 5, Ocn$^{-/-}$: n = 6), female (n = 9, 6) at 11w; male (n = 25, 18) at 14w; male (n = 14, 23), female (n = 6, 8) at 6m; male (n = 7, 9), female (n = 8, 12) at 9m; female (n = 8, 6) at 12m; male (n = 7, 4) at 18m. (B) Blood glucose levels. Male (wt: n = 5, Ocn$^{-/-}$: n = 6) at 14w; male (n = 16, 16), female (n = 8, 13) at 9m; male (n = 7, 5) at 18m. (C) HbA1c levels. Male (wt: n = 5, Ocn$^{-/-}$: n = 6), female (n = 9, 10) at 11w; male (n = 14, 18), female (n = 6, 7) at 6m; male (n = 9, 7), female (n = 9, 13) at 9m; male (n = 7, 5) at 18m. (D-H) Measurement of adipose tissues. D-G, Sagittal sections at the midline (D, E) and cross sections at the level of L5 (F, G) of μ-CT in male mice at 14 weeks of age. Subcutaneous adipose tissues are shown orange and visceral adipose tissues are shown yellow in the range of the first to fifth vertebra. H, Volumes of subcutaneous and visceral adipose tissues in wild-type (n = 10) and Ocn$^{-/-}$ (n = 6) male mice at 14 weeks of age and wild-type (n = 8) and Ocn$^{-/-}$ (n = 5) female mice at 9 months of age. The percentages of adipose tissue volumes against abdominal volumes in the range of the first to fifth vertebra are shown. X symbols in box plots show outliers.

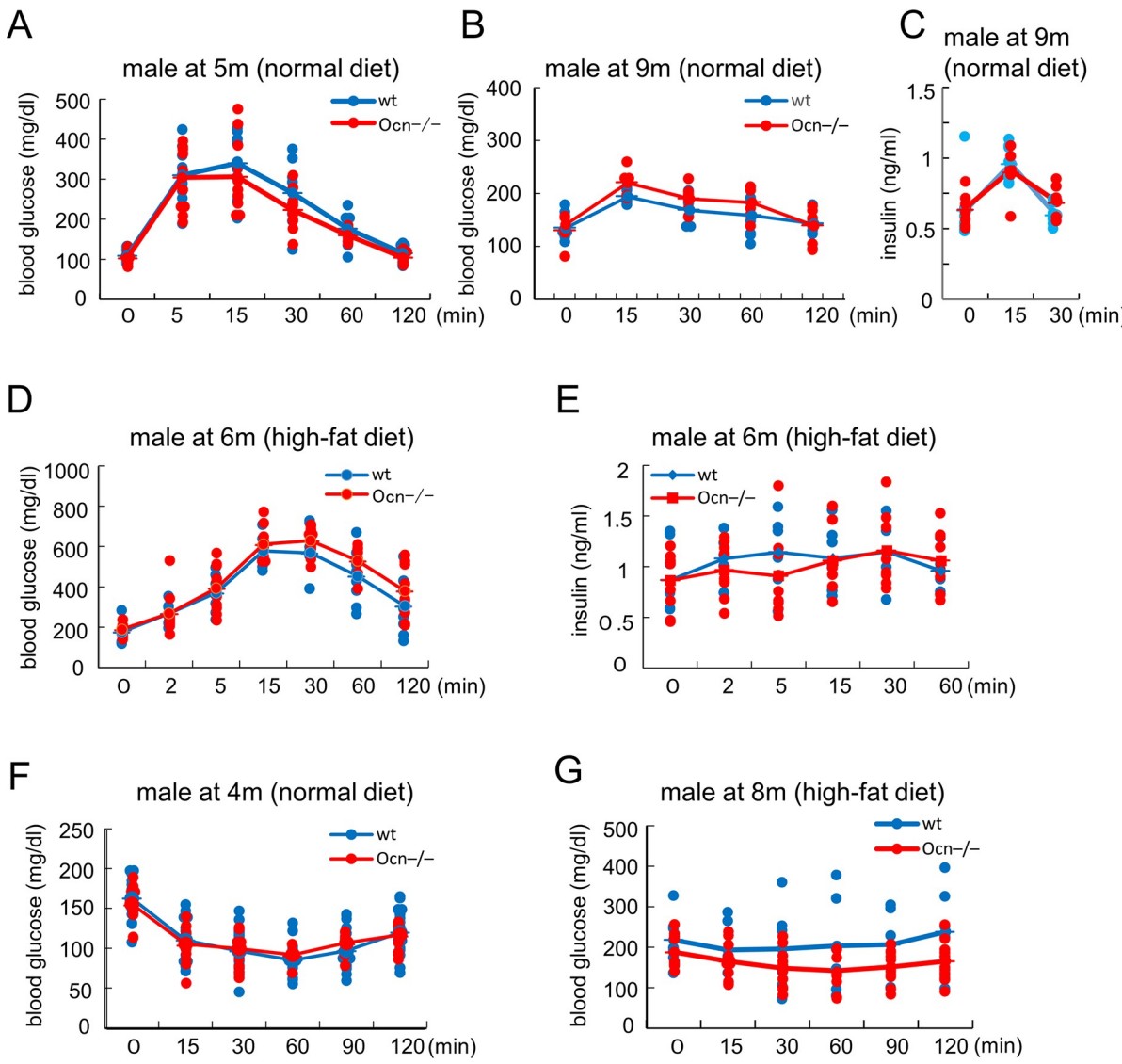

**Fig 9. GTTs and ITTs in Ocn$^{-/-}$ mice.** (A) GTT by glucose (2 g/kg body weight) injection. Glucose levels in male mice (wt: n = 9, Ocn$^{-/-}$: n = 12) at 5 months of age fed a normal diet. (B and C) GTT by glucose (1 g/kg body weight) injection. Glucose (B) and insulin (C) levels in male mice at 9 months of age fed a normal diet. wt: n = 9, Ocn$^{-/-}$: n = 7. (D and E) GTT by glucose (2 g/kg body weight) injection. Glucose (D) and insulin (E) levels in male mice (wt: n = 8, Ocn$^{-/-}$: n = 12) at 6 months of age fed a high-fat diet for 1 month. (F) ITT. Glucose levels in male mice (wt: n = 20, Ocn$^{-/-}$: n = 12) at 4 months of age fed a normal diet. (G) ITT. Glucose levels in male mice (wt: n = 7, Ocn$^{-/-}$: n = 12) at 8 months of age fed a high-fat diet for 3 months.

despite the fact that orientation of collagen along the same axis remained unchanged, leading to the deterioration of mechanical properties. These results demonstrate that Ocn plays a role in the epitaxial growth of BAp on the collagen template and is essential for ensuring that the directionality of the BAp $c$-axis is parallel to collagen fibrils, which run longitudinally to bone. Thus, Ocn plays an important role in maintaining bone quality and strength by adjusting the crystallographic orientation of the BAp $c$-axis parallel to collagen fibrils. Random blood glucose and HbA1c levels, the amounts of subcutaneous and visceral adipose tissues, glucose and insulin levels in GTTs, and glucose levels in ITTs in Ocn$^{-/-}$ mice were similar to those of wild-type mice in both sexes fed a normal or high-fat diet and at all ages examined. Further, testicular weights, serum testosterone levels, the numbers of spermatozoa and TUNEL-positive germ

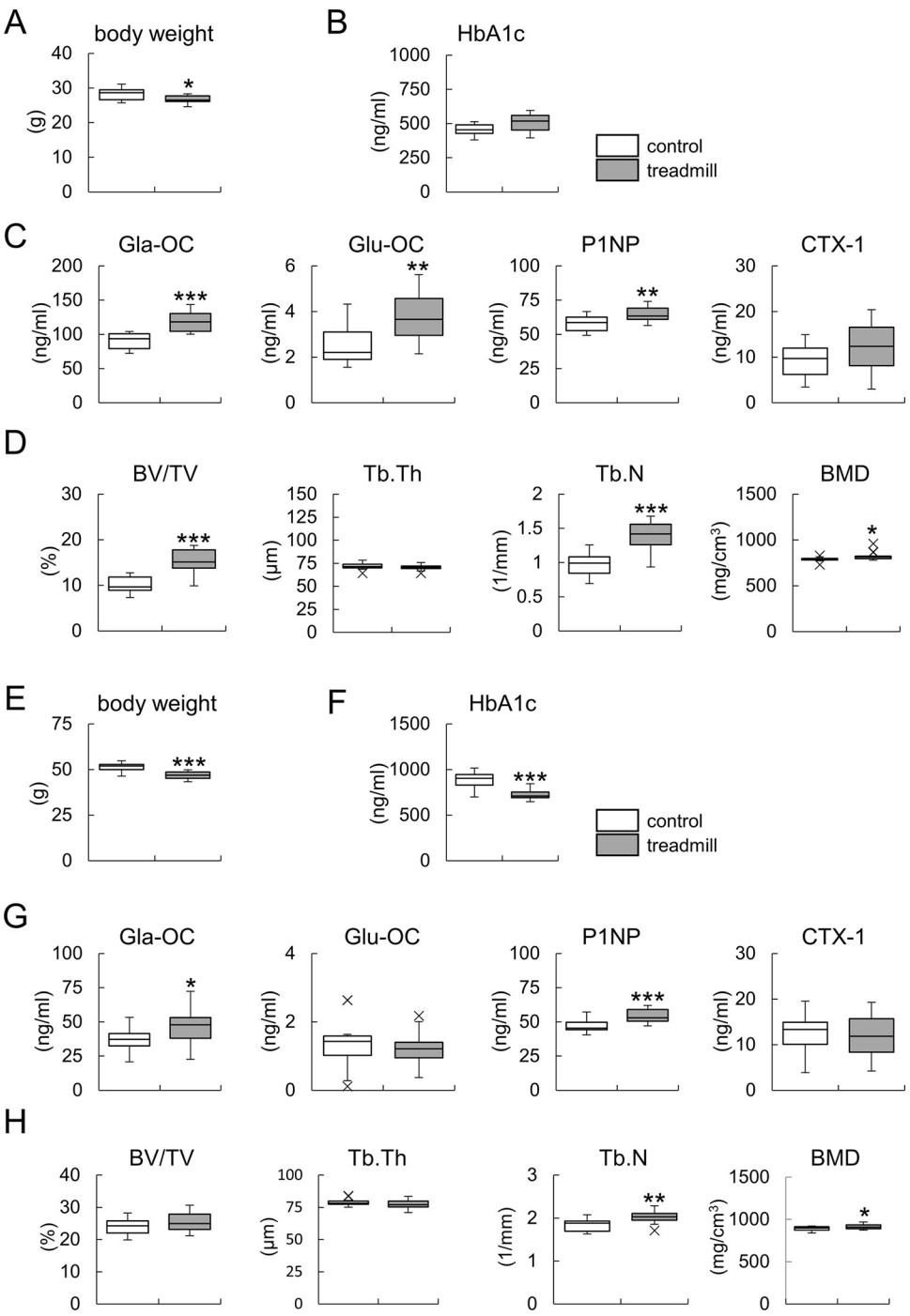

**Fig 10. Relationships between exercise, glucose metabolism, and bone formation.** (A-D) Body weights (A), HbA1c (B), serum levels of carboxyleted Ocn (Gla-OC), uncarboxyleted Ocn (Glu-OC), P1NP, and CTX-1 (C), and μ-CT analysis of trabecular bone (D) in male wild-type mice fed a normal diet with or without exercise on a treadmill for 7 weeks. Mice were analyzed at 4 months of age. n = 13. (E-H) Body weights (E), HbA1c (F), serum levels of carboxyleted Ocn, uncarboxyleted Ocn, P1NP, and CTX-1 (G), and μ-CT analysis of trabecular bone (H) in male KK/TaJcl mice with or without exercise on a treadmill for 7 weeks. Mice were fed a high-fat diet and analyzed at 4 months of age. Control: n = 15, treadmill: n = 14. *: P<0.05, **: P<0.01, ***: P<0.001. X symbols in box plots show outliers.

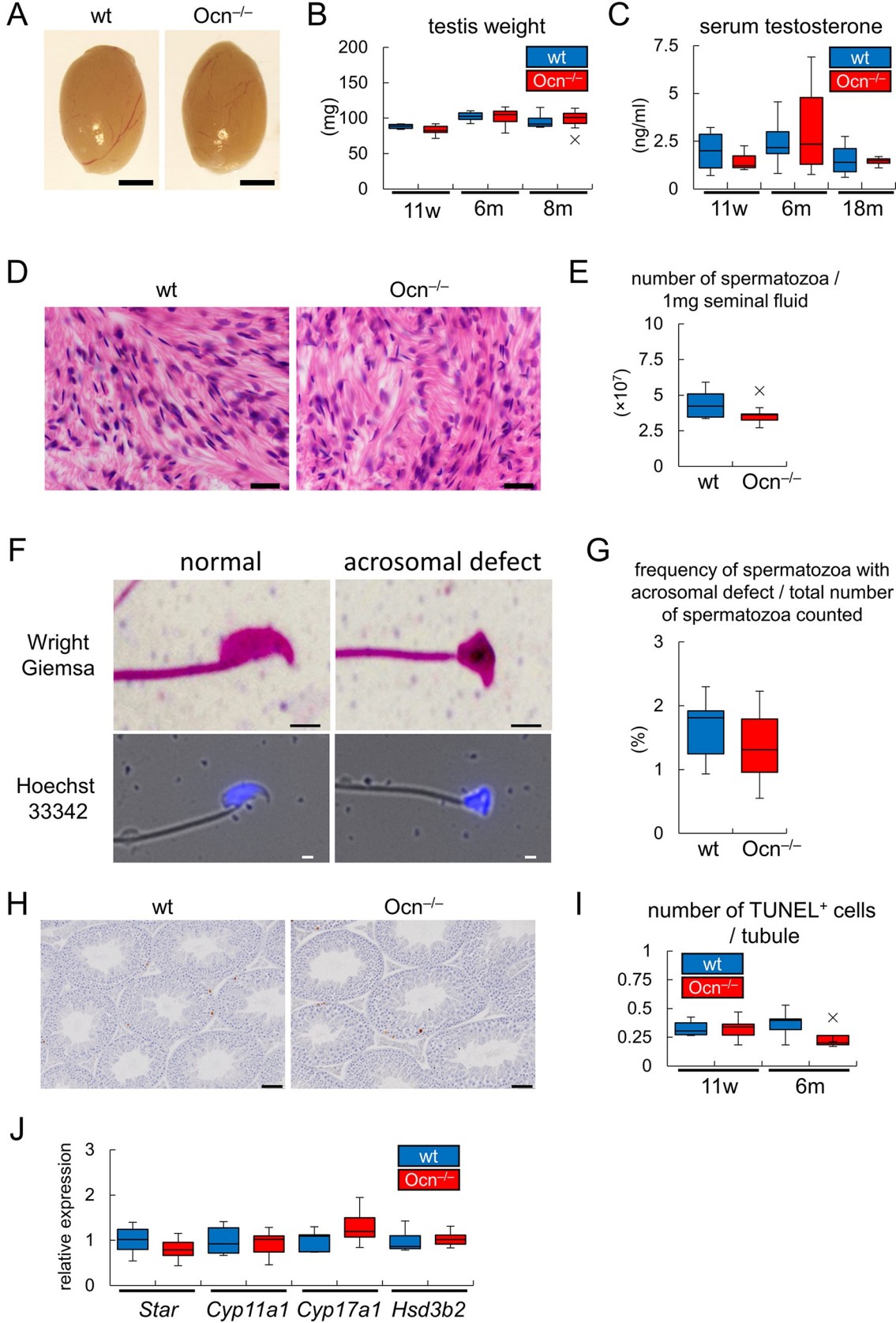

**Fig 11. Analyses of testosterone synthesis and spermatogenesis.** (A and B) Appearance of testes (A) and their weights (B) in wild-type (n = 5 at 11w, n = 7 at 6m, n = 7 at 8m) and Ocn$^{-/-}$ (n = 6 at 11w, n = 7 at 6m, n = 11 at 8m) mice. Bars: 2 mm. (C) Serum testosterone levels in wild-type (n = 4 at 11w, n = 12 at 6m, n = 6 at 18m) and Ocn$^{-/-}$ (n = 3 at 11w, n = 15 at 6m, n = 4 at 18m) mice. (D) H-E stained sections of the cauda epididymides at 4 months of age. Bars: 10 μm. (E) The number of spermatozoa. The number of spermatozoa in 1 mg of seminal fluid of each mouse at 8 months of age are shown. wt: n = 7, Ocn$^{-/-}$: n = 11. (F) Morphology of sperms. Sperms were stained with Wright-Giemsa or analyzed with fluorescence microscope after the treatment with Hoechst 33342. Bars: 2 μm. (G) The frequencies of the sperms with acrosomal defects. More than 300 sperms in each mouse at 8 months of age were evaluated using the smears stained with Wright-Giemsa. wt: n = 7, Ocn$^{-/-}$: n = 11. (H) TUNEL staining. The sections were counterstained with hematoxylin. Bars: 50 μm. (I) The number of TUNEL-positive cells in seminiferous tubules. All tubules (more than 200 tubules) in one section were evaluated in each mouse at 11 weeks of age (wt: n = 5, Ocn$^{-/-}$: n = 6) and 6 months of age (wt: n = 5, Ocn$^{-/-}$: n = 4), and the means of the number of TUNEL-positive cells in one tubule are shown. (J) Real-time RT-PCR analysis using RNA from testes at 8 months of age. The values in wild-type mice were defined as 1, and relative levels are shown. wt: n = 5, Ocn$^{-/-}$: n = 8. X symbols in box plots show outliers.

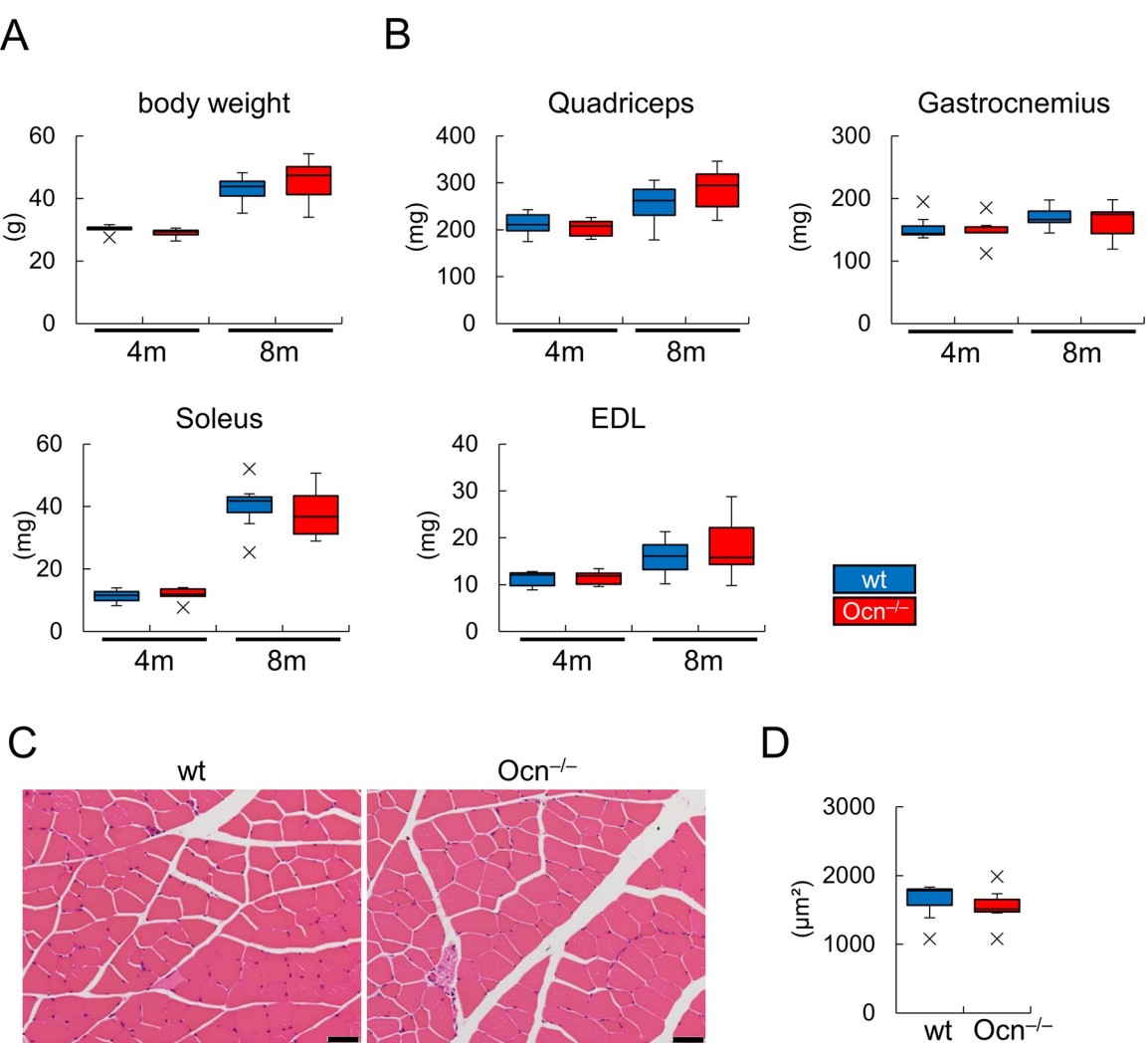

**Fig 12. Analyses of muscles.** (A and B) Body weights (A) and muscle weights (B) in male mice at 4 months (wt: n = 5, Ocn$^{-/-}$: n = 6) and 8 months (wt: n = 7, Ocn$^{-/-}$: n = 11) of age. (C and D) H-E stained sections of the quadriceps (C), and the average area of myofibers in the quadriceps (D) of male wild-type and Ocn$^{-/-}$ mice at 4 months of age. The areas of sixty myofibers in the cross sections at mid-diaphyses of femurs were measured. wt: n = 7, Ocn$^{-/-}$: n = 7. EDL: extensor digitorum longus. Bars: 50 μm. X symbols in box plots show outliers.

cells, as well as the expression of the genes encoding the enzymes necessary for testosterone biosynthesis were normal in Ocn$^{-/-}$ mice. Moreover, muscle mass was normal in Ocn$^{-/-}$ mice. These results indicate that osteocalcin is not physiologically involved in glucose metabolism, testosterone synthesis, or the maintenance of muscle mass.

In normal mineralization, the *c*-axis of apatite, which is the predominant direction of crystal growth, aligns almost parallel to the direction of collagen fibrils, which preferentially run along the longitudinal axis of long bones. This is due to the epitaxial crystallization of apatite on the collagen template [43]. Immunolocalization clarified that Ocn exists in intrafibrillar and interfibrillar collagen regions (Fig 4A, 4B, 4D and 4E), whereas another non-collagenous protein, bone sialoprotein, is only present in interfibrillar collagen regions in mineralized tissue [44]. These findings suggest the involvement of Ocn in the bone mineralization process *in vivo*. Poundarik et al. recently reported that Ocn$^{-/-}$ mice have smaller and less organized BAp particles along the longitudinal axis of femurs by a SAXS analysis, which provides information on the shape of bone apatite particles, but not the crystallographic orientation (atomic arrangement) [13].

The orientation of collagen has not yet been examined [13]. Hence, the directional relationship between collagen and BAp remains unclear [13]. The present results revealed that collagen fibers preferentially oriented along the longitudinal axis in Ocn$^{-/-}$ femurs with an identical orientation degree to wild-type femurs (Fig 6B and 6D). Furthermore, the results of the μXRD analysis showed the severely disrupted orientation of the BAp *c*-axis in Ocn$^{-/-}$ femurs (Fig 6E). Therefore, our results reveal for the first time the disruption of the crystallographic orientation of BAp in Ocn$^{-/-}$ bone, in spite of the normal orientation of collagen fibers.

The orientations of collagen and the BAp *c*-axis have been investigated in several mouse models and bone types, including *oim/oim* osteogenesis imperfecta, c-src$^{-/-}$ osteopetrosis, melanoma metastases, unloading, and regenerating long bones [40,45–48]. Different from the Ocn$^{-/-}$ bone, the orientation of collagen was disrupted and, as a result, the orientation of the BAp *c*-axis along the bone longitudinal axis was severely disrupted. In these bones, an epitaxial relationship still exists between collagen and BAp. Thus, ours is the first study to show that the orientation of the BAp *c*-axis was disrupted even though collagen fibrils ran in the correct direction (longitudinal axis of the long bone). These results demonstrate that Ocn is essential for the alignment of the BAp *c*-axis to the direction of collagen fibrils.

The present study also shows that the *c*-axis of apatite crystallites is not always aligned synchronously along the orientation of collagen and, therefore, collagen is not the only factor influencing the orientation of apatite, which is consistent with the findings of Glimcher [49]. According to biomechanics, the orientations of collagen and apatite contribute differently to bone mechanical properties; highly oriented collagen mainly contributes to the prevention of crack propagation along the vertical direction against the collagen running axis [50], while highly oriented apatite *c*-axes predominantly contribute to stiffening the bone matrix [48]. Individual analyses on the orientations of collagen and apatite are needed to understand the functions of bone, clarify the pathology of bone diseases, and select optimal anti-bone disease agents. Furthermore, Young's modulus in this work strongly correlated with the orientation of the BAp *c*-axis, but not that of collagen or BMD. These results indicate that the reduction in Young's modulus was due to the disrupted alignment of the BAp *c*-axis to the longitudinal axis of bone. This evidence demonstrates that the orientation of the BAp *c*-axis to the longitudinal axis of bone by osteocalcin is important for bone strength.

A schematic view of the directional relationship between collagen fibrils and the BAp *c*-axis in wild-type and Ocn$^{-/-}$ mice is depicted in Fig 7D. Collagen fibrils and the BAp *c*-axis co-align parallel to the longitudinal axis in normal bone, whereas the orientation of the BAp *c*-axis is severely disrupted in spite of normally oriented collagen fibrils parallel to the bone

longitudinal axis in Ocn$^{-/-}$ bone, leading to a reduction in Young's modulus in the longitudinal direction. To clarify how Ocn aligns the BAp $c$-axis to the direction of collagen fibrils, interactions between Ocn and collagen and between Ocn and BAp need to be investigated. The binding potential of Ocn to Ca$^{2+}$ in the (001) crystallographic plane of apatite lattices [51] is considered to be a key in BAp growth toward a specific direction along the orientation of collagen. Regarding the directional (epitaxial) relationship, a simulation study of ionic clustering behavior—the very early stage of BAp nucleation—under an interaction with affector molecules may provide an answer [52].

Some clinical studies showed a positive relationship between total Ocn and glucose metabolism [53–58]; however, the role of uncarboxyleted Ocn in glucose metabolism remains controversial [15–24]. Exercise reduced body weight and HbA1c and increased serum Ocn, P1NP, and BMD in KK/TaJcl mice (Fig 10E–10H), suggesting that exercise improves glucose metabolism and increases serum Ocn as a result of increased bone formation. In studies showing a relationship between serum total Ocn and glucose metabolism, the serum levels of bone-specific alkaline phosphatase, which is a marker for bone formation, were elevated [59,60], suggesting that the increase in serum total Ocn was due to enhanced bone formation. Thus, the relationship between Ocn and glucose metabolism may be accounted for by the exercise induced change in bone formation.

The previously reported phenotypes of increased bone mass, impaired glucose metabolism, and reduced testosterone synthesis and muscle mass in the Karsenty Ocn$^{-/-}$ mice were not observed in our Ocn$^{-/-}$ mice, in spite of the fact that both *Bglap* and *Bglap2* were deleted and *Bglap3* was intact in both Ocn$^{-/-}$ mouse lines. To the best of our knowledge, the only difference between the two mouse lines is the genetic background. Our Ocn$^{-/-}$ mice were backcrossed to C57BL/6N more than 8 times before analysis, while the previous Ocn$^{-/-}$ mice were analyzed in the mixed background of C57BL/6J and 129/SV. Since the difference of the genetic background is unlikely to explain all controversial phenotypes, our Ocn$^{-/-}$ mouse line will be shared with other researchers to resolve the controversies.

In conclusion, Ocn is not necessary for the regulation of bone formation or resorption, but is essential for the alignment of the BAp $c$-axis parallel to collagen fibrils and is required for optimal bone strength. Moreover, Ocn does not work as a hormone that regulates glucose metabolism, testosterone synthesis, and muscle mass.

## Methods

### Generation of Ocn$^{-/-}$ mice

The targeting vector for the mouse *Bglap* and *Bglap2* genes was generated by the BAC modification technique as previously described [61]. The BAC modification kit was purchased from Gene Bridges (Heidelberg, Germany), and the BAC clone (RF24-298F19) was purchased from the BAC-PAC Resource Center (Oakland, CA). The cassette containing PGK-gb2-neo was PCR amplified and introduced into bacteria containing the BAC clone to replace genomic DNA encompassing *Bglap* and *Bglap2*. The cassette containing PGK-gb2-neo flanked by 5' DNA of *Bglap* and 3' DNA of *Bglap2* was cloned into the plasmid containing the PGK-tk cassette. The final targeting construct was electroporated into the E14 line of ES cells, and targeted ES cells were injected into C57BL/6 blastocysts. The mutant allele was confirmed by a Southern blot analysis as described previously [62]. Ocn$^{+/-}$ mice were backcrossed with C57BL/6N 8–10 times. Prior to the study, all experiments were reviewed and approved by the Animal Care and Use Committee of Nagasaki University Graduate School of Biomedical Sciences (No. 1903131520–2). Animals were housed 3 per cage in a pathogen-free environment on a 12-h light cycle at 22±2˚C with standard chow (CLEA Japan, Tokyo) and had free access to tap water.

### Real-time RT-PCR analysis

Total RNA was extracted using ISOGEN (Wako, Osaka, Japan). Osteoblast-enriched cells were collected as previously described [63]. Real-time RT-PCR was performed on RNA isolated from the femurs or testes of individual mouse using the primers shown in S2 Table as previously described [64]. We normalized the values obtained to those of *β-actin*.

### μ-CT, bone histomorphometry, and TEM

Dissected femurs were analyzed by μ-CT systems (R_mCT; Rigaku Corporation, Tokyo, Japan) as previously described [63]. The volume of adipose tissue was analyzed by the μ-CT system using 3D fat analysis software (Rigaku Corporation). Bone histomorphometric analyses were performed as previously described [65]. In the assessment of dynamic histomorphometric indices, mice were injected with calcein 6 and 2 d before sacrifice at a dose of 0.16 mg/10 g body weight. In the immunoelectron microscopic analysis, mice were perfusion-fixed with 0.01% glutaraldehyde-4% paraformaldehyde in 0.05 M cacodylate buffer (pH7.4). The dissected long bones were further immersed in the same fixatives at 4°C for 24 h, decalcified in 5% EDTA (pH7.4) at 4°C for 2–4 weeks, dehydrated with ascending concentrations of ethanol, and embedded in LR-white resin (London Resin Company Ltd., Berkshire, UK). Ultrathin sections (90 nm) were immunolabeled with a rabbit anti-Ocn antibody (Takara Bio Inc., Shiga, Japan), followed by a 20-nm colloidal gold-conjugated secondary antibody to Rabbit IgG (BBI Solutions, Cardiff, UK), and post-stained with 2% uranyl acetate. In the TEM analysis of non-decalcified specimens, mice were perfusion-fixed with 2.5% glutaraldehyde-2% paraformaldehyde in 0.1 M cacodylate buffer (pH7.4). The dissected long bones were further immersed in the same fixatives at 4°C for 24 h, post-fixed with 1% $OsO_4$ in 0.1 M cacodylate buffer (pH7.4) at 4°C for 4 h, dehydrated with ascending concentrations of acetone, and embedded in epoxy resin (Taab, Berkshire, UK).

### Assays for Ocn, markers for bone formation and resorption, and testosterone

The serum level of total Ocn was examined using the Ocn EIA kit (BTI Biomedical Technologies, Inc., Stoughton. MA), that of carboxyleted Ocn by the Mouse Gla-Osteocalcin High Sensitive EIA Kit (Takara Bio Inc., Shiga Japan), that of uncarboxyleted Ocn by the Mouse Glu-Osteocalcin High Sensitive EIA Kit (Takara Bio Inc.), that of P1NP by Rat/Mouse P1NP ELISA (Immunodiagnostic Systems, Boldon, UK), that of TRAP5b by the Mouse TRAP Assay (Immunodiagnostic Systems), and that of CTX1 by RatLaps (CTX-I) EIA (Immunodiagnostic Systems). Serum testosterone levels were assessed using Testosterone Rat/Mouse ELISA (Demeditec Diagnostics GmbH, Kiel, Germany).

### Confocal Raman spectroscopy

The area (80 × 60 μm) adjacent to the endosteal surface of the posterior side in slices of the mid-diaphysis of femurs were analyzed by a confocal Raman microscope (Nanofinder HE, Tokyo Instruments Inc., Japan). A 532-nm laser was focused on 200 fields in the area with 2-μm spot sizes. Each field was exposed for 0.1 sec twice, and Raman spectra was obtained by totalizing 400 values in each sample.

### Analysis of BMD and collagen orientation

Volumetric BMD was measured at 9 points (regular intervals of 1/10 of the bone length along the longitudinal axis) on femurs at 9 months of age using an XCT Research SA+ system

(STRATEC Medizintechnik Gmbh, Birkenfeld, Germany) with a resolution of $70 \times 70 \times 260$ μm as described previously [40]. The femurs at 9 months of age were decalcified and embedded in paraffin, and sagittal sections (10 μm) were deparaffinized and observed using a two-dimensional birefringence analyzer (WPA-micro, Photonic Lattice, Miyagi, Japan). Collagen orientation was assessed at 9 points in the anterior and posterior sides of cortical bone in femurs as previously described [40]. To evaluate the collagen orientation degree along the longitudinal direction of the femur, the orientation order parameter $f_\theta$ was calculated based on the angle distribution of collagen against the bone longitudinal direction as previously described [66]. $f_\theta$ takes a value ranging from -1 (collagen perfectly aligned perpendicular to the bone longitudinal direction) to 1 (collagen perfectly aligned parallel to the bone longitudinal direction).

## Analysis of the BAp *c*-axis orientation

The crystallographic orientation of apatite crystallites was analyzed by the μXRD system (R-Axis BQ, Rigaku, Tokyo, Japan), which is equipped with a transmission-type optical system and an imaging plate (storage phosphors) (Fuji Film, Tokyo, Japan), at 9 points (regular intervals of 1/10 of the bone length along the longitudinal axis) of cortical bone in femurs as previously described [40] with a minor modification in that the incident beam was focused on a beam spot of 300 μm in diameter and diffraction data were collected for 180 sec. The preferential orientation degree of the BAp *c*-axis was assessed as the relative intensity ratio of the (002) diffraction peak to the (310) peak in the X-ray profile.

## Analysis of the size of BAp crystallites

The average size of BAp crystallites in the *c*-axis was calculated based on the Scherrer formula using the half width of the diffraction peak profile of (002) along the longitudinal and tangential axes of bone analyzed by the microbeam X-ray diffractometer system (D8 DISCOVER with GADDS, Bruker AXS, WI, USA). Cu-Kα radiation with a wavelength of 0.15418 nm was generated at a tube voltage of 45 kV and a tube current of 110 mA. An incident beam was focused on a beam spot of 100 μm and diffraction data were counted with a two-dimensional position sensitive proportional counter (PSPC) for 600 sec. The half width of the observed diffraction profile was corrected by that of hydroxyapatite (NIST #2910: calcium hydroxyapatite) to acquire the half width of the pure profile.

## Nanoindentation testing

Young's modulus was measured along the longitudinal direction of the femur using midshaft specimens (position 5) by a nanoindentation system (ENT-1100a; Elionix, Tokyo, Japan) with a Berkovich diamond indenter as previously described [40]. Young's modulus was assessed according to the method of Oliver and Pharr [67].

## Metabolic assessments

Regarding GTTs, glucose (1 or 2 g/kg body weight) was injected intraperitoneally after fasting for 15 hours. Regarding ITTs, insulin (1 U/kg body weight) was injected intraperitoneally after fasting for 6 hours. Blood glucose was measured using Glutest Neo Super (Sanwa Kagaku Kenkyusho Co., Ltd., Nagoya Japan). Serum insulin was measured using the Mouse Insulin ELISA Kit (Morinaga Institute of Biological Science, Inc., Yokohama, Japan). HbA1c was measured using the Mouse HbA1c ELISA Kit (Aviva Systems Biology, San Diego, CA). Mice were fed a

normal diet (CE-2, CLEA Japan Inc., Tokyo, Japan) or high-fat diet (HFD-32, CLEA Japan Inc.).

### Exercise on a treadmill

Wild-type (C57BL/6) mice were fed a normal diet, and the mouse model for type 2 diabetes mellitus, KK/TaJcl (CLEA Japan Inc.), was fed a normal diet until one week before the exercise study and then fed a high-fat diet. Mice were divided into two groups with or without exercise. Mice were forced to exercise on a treadmill (Natsume Seisakusho Co., Ltd., Tokyo, Japan) with an electric shock at 20V/50mA to trigger running. After running training for 3 days, mice ran for 30 min per day at a speed of 12 m/min on the treadmill with a tilt of 5 degrees. Running continued 5 days per week for 7 weeks. Blood sampling was performed the day after the last exercise session.

### Histological analysis

Mice were perfusion-fixed with 4% paraformaldehyde/0.01 M phosphate-buffered saline. Testes with epididymides and muscle were embedded in paraffin. Sections (thickness of 3 μm) were stained with hematoxylin and eosin (H-E) or stained for TUNEL using the ApopTag Peroxidase In Situ Apoptosis Detection Kit (Sigma Aldrich, St. Louis, MO).

### Analysis of sperm

Seminal fluid was collected from the cauda epididymides of male mice at 8 months of age and the weight was measured. The sperm were incubated with the medium (FERTIUP Kyudo Co. Ltd. Saga, Japan) containing Hoechst 33348 (1.5μg/ml, Sigma Aldrich) for 1 hour. After incubation, the medium containing sperm was diluted with PBS and the number of sperm was counted. For analysis of sperm morphology, the sperm was observed by fluorescence microscope (All-in-one Fluorescence Microscope BX-X710, KEYENCE, Osaka, Japan). Further, the diluted medium containing sperm was smeared to slide glasses and the smears were stained with Wright-Giemsa and analyzed with a Zeiss Axioskop 2 plus microscope (Carl Zeiss, Tokyo, Japan) with a Olympus DP74 camera (Olympus Co., Tokyo, Japan).

### Box plots and statistical analysis

Data were shown using the box plots to visually show the distribution of numerical data and skewness by displaying the data quartiles and averages. The outliers were defined as data points located outside 1.5 times the interquartile range (IQR, shown as a box) above the upper quartile or those located outside 1.5 times the IQR below the lower quartile. Statistical analyses were performed using the Student's $t$-test for the comparison between wild-type and $Ocn^{-/-}$ mice. A P-value of less than 0.05 was considered to be significant. The statistical analyses and single and multiple regression analyses were performed using BellCurve for Excel (Social Survey Research Information Co., Ltd., Tokyo, Japan).

### Supporting information

**S1 Fig. μ-CT analysis of femurs in female wild-type and $Ocn^{-/-}$ mice at 6 and 9 months of age.** (A and B). μ-CT images of femoral distal metaphyses (A) and mid-diaphyses (B). Scale bars = 500 μm. (C) Trabecular bone parameters, including the trabecular bone volume (BV/TV), trabecular thickness (Tb.Th), and trabecular number (Tb.N). (D) Cortical bone parameters including cortical thickness (Ct. Th), the periosteal perimeter (Ps.Pm), and endocortical perimeter (Ec.Pm). wt (n = 6), $Ocn^{-/-}$ (n = 8) at 6m; wt (n = 5), $Ocn^{-/-}$ (n = 6) at 9m. X symbols

in box plots show outliers. (TIF)
(TIF)

**S2 Fig. μ-CT analysis and serum markers after ovariectomy.** Sham operation or ovariectomy (OVX) was performed at 5 weeks of age and mice were analyzed at 11 weeks of age. (A-D) μ-CT analyses of femurs in wild-type and Ocn$^{-/-}$ mice with sham operation or OVX. μ-CT images of femoral distal metaphyses (A) and mid-diaphyses (B) are shown. Scale bars = 500 μm. C, Trabecular bone parameters, including the trabecular bone volume (BV/TV), trabecular thickness (Tb.Th), and trabecular number (Tb.N). D, Cortical bone parameters including cortical thickness (Ct. Th), the periosteal perimeter (Ps.Pm), and endocortical perimeter (Ec.Pm). wt sham: n = 9, wt OVX: n = 8, Ocn$^{-/-}$ sham: n = 6, Ocn$^{-/-}$ OVX: n = 7. X symbols in box plots show outliers. (TIF)
(TIF)

**S3 Fig. H-E-stained sections of wild-type and Ocn$^{-/-}$ femurs at 14 weeks of age.** (A, C, E) Wild-type mice. (B, D, F) Ocn$^{-/-}$ mice. The boxed regions in A are magnified in C and E, and those in B are magnified in D and F. Scale bars: 0.5 mm (A, B), 200 μm (C, D), 50 μm (E, F). (TIF)
(TIF)

**S4 Fig. Histological analysis of osteoclasts.** Sections of femurs in wild-type (A, C) and Ocn$^{-/-}$ (B, D) mice at 14 weeks of age were stained with TRAP. The boxed regions in A and B are magnified in C and D, respectively. Scale bars: 200 μm (A, B), 100 μm (C, D). (TIF)
(TIF)

**S5 Fig. BMD, orientations of collagen fibers and the BAp $c$-axis, and Young' modulus in the femoral cortical bone of male wild-type and Ocn$^{-/-}$ mice at 14 weeks of age.** (A) Schematic presentation of analyzed positions. (B) BMD at position 5. (C) Collagen orientation degree. (D) BAp $c$-axis orientation degree. (E) Single regression analysis of the orientations of collagen fibers and the BAp $c$-axis. (F) Young's modulus along the bone longitudinal axis at position 5. $^{**}$: P<0.01. X symbols in box plots show outliers. (G) Single regression analysis of Young's modulus to BMD and each degree of the preferential alignment of collagen fibers and the BAp $c$-axis in the bone longitudinal direction at position 5. wt: blue dots, Ocn$^{-/-}$: red dots. n = 6. (TIF)
(TIF)

**S6 Fig. The three-point bending test.** Representative load-displacement curves for male wild-type (A) and Ocn$^{-/-}$ (B) mice at 6 months of age, in which maximum load, displacement, stiffness (the slope of the linear part of the load), and energy to failure (area under the load-displacement curve) were obtained. C, Maximum load, displacement, stiffness, and energy in wild-type (n = 8) and Ocn$^{-/-}$ (n = 6) mice. The experiments were performed as previously described (J Bone Miner Res 2016; 31: 1366–1380.).(TIF)
(TIF)

**S7 Fig. GTTs.** Glucose (1 g/kg body weight) was injected intraperitoneally in GTTs. (A) Glucose levels in male mice at 14 weeks of age fed a normal diet. wt: n = 5, Ocn$^{-/-}$: n = 6. (B) Glucose levels in female mice at 9 months of age fed a normal diet. wt: n = 4, Ocn$^{-/-}$: n = 6. (C and D) Glucose (C) and insulin (D) levels in male mice at 18 months of age fed a normal diet. wt: n = 7, Ocn$^{-/-}$: n = 5. (E) Glucose levels in male mice at 14 weeks of age fed a high-fat diet for 5 weeks. wt: n = 5, Ocn$^{-/-}$: n = 4. (F) Glucose levels in female mice at 14 weeks of age fed a high-fat diet for 5 weeks. wt: n = 7, Ocn$^{-/-}$: n = 5. (G) Glucose levels in female mice at 9 months of

age fed a high-fat diet for 11 weeks. wt: n = 4, Ocn$^{-/-}$: n = 5. (TIF)
(TIF)

**S8 Fig. μ-CT analysis of cortical bone.** Male wild-type (A) and KK/TaJcl (B) mice with or without exercise on a treadmill for 7 weeks were analyzed by μ-CT at 4 months of age. Cortical thickness (Ct. Th), the periosteal perimeter (Ps.Pm), endocortical perimeter (Ec.Pm), and BMD are shown. Control: n = 13, treadmill: n = 13 in wild-type mice. Control: n = 15, treadmill: n = 14 in KK/TaJcl mice. X symbols in box plots show outliers. (TIF)
(TIF)

**S9 Fig. Analyses of spermatogenesis and muscle.** (A-J) Histological sections of testis and epididymis at 4 months of age stained with H-E. Testis and epididymis at low magnification (A, B), seminiferous tubules (C-F), and cauda epididymis (G-J) are shown. Boxed regions in C, D, G, and H are magnified in E, F, I, and J, respectively. Boxed regions in I and J are magnified in Fig 11D. Bars: 1mm (A, B); 100 μm (C, D, G, H); and 20 μm (E, F, I, J). (K-N) Muscle weights (K), H-E stained sections (L, M), and the average areas of myofibers (N) in gastrocnemius muscle in male wild-type (n = 8) and Ocn$^{-/-}$ (n = 7) mice at 9 months of age. Bars: 50 μm. X symbols in box plots show outliers. (TIF)
(TIF)

**S1 Table. Multiple regression analysis using the BAp *c*-axis orientation, collagen orientation, and BMD as explanatory variables and Young's modulus as a response variable at 14 weeks of age.** β: standard partial regression coefficient.
(XLSX)

**S2 Table. Primer sequences.**
(XLSX)

**S1 Data. Spreadsheets with numerical data used to generate all Figures.**
(XLSX)

## Acknowledgments

We thank H. Sano for technical assistant in confocal Raman spectroscopy and S. Hamachi for secretarial assistance.

## Author Contributions

**Formal analysis:** Takeshi Moriishi, Ryosuke Ozasa, Takuya Ishimoto, Takayoshi Nakano, Ryo Fukuyama.

**Funding acquisition:** Takeshi Moriishi, Toshihisa Komori.

**Investigation:** Takeshi Moriishi, Ryosuke Ozasa, Takuya Ishimoto, Takayoshi Nakano, Tomoka Hasegawa, Toshihiro Miyazaki, Wenguang Liu, Ryo Fukuyama, Yuying Wang, Hisato Komori, Xin Qin, Toshihisa Komori.

**Methodology:** Takeshi Moriishi, Ryosuke Ozasa, Takuya Ishimoto, Takayoshi Nakano, Tomoka Hasegawa, Toshihiro Miyazaki, Wenguang Liu, Norio Amizuka.

**Project administration:** Takeshi Moriishi, Toshihisa Komori.

**Supervision:** Toshihisa Komori.

**Validation:** Takeshi Moriishi, Takayoshi Nakano, Norio Amizuka, Toshihisa Komori.

**Visualization:** Takeshi Moriishi, Takayoshi Nakano, Toshihisa Komori.

**Writing – original draft:** Toshihisa Komori.

**Writing – review & editing:** Takayoshi Nakano, Toshihisa Komori.

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
