## [Decision Letter · Decision Letter 0]

1 Oct 2019

Dear Dr Komori,

Thank you very much for submitting your Research Article entitled 'Osteocalcin is necessary for the alignment of apatite crystallites, but not for regulating glucose metabolism, testosterone synthesis, or muscle mass' to PLOS Genetics. Your manuscript was fully evaluated at the editorial level and by independent peer reviewers. The reviewers appreciated the attention to an important topic but identified some aspects of the manuscript that should be improved.

We therefore ask you to modify the manuscript according to the review recommendations before we can consider your manuscript for acceptance. Your revisions should address the specific points made by each reviewer.

[LINK]

Yours sincerely,

John F Bateman

Associate Editor

PLOS Genetics

Gregory Barsh

Editor-in-Chief

PLOS Genetics

This manuscripts reports studies on a new mouse of osteocalcin deficiency which challenges early findings on another osteocalcin mouse model, thus providing new information important for bone and mineral metabolism research

The external reviewers suggested minor revisions which will provide further clarity and improve readability. We would expect that these concerns can be readily addressed by the authors in a revised manuscript.

One of the reviewers made suggestions on some rewording of the title, abstract and author summary. The editors do not insist on these specific editorial changes, but only that the authors consider these as general comments when providing a revised manuscript.

Reviewer's Responses to Questions

**Comments to the Authors:**

Reviewer #1: During the last twenty three years, the bone and mineral metabolism research community (and the NIH institutes and numerous other national and international funding agencies that support it) have devoted a great deal of intellectual energy and resources on some very provocative, albeit controversial, ideas about osteocalcin (OCN) – a 46 amino-acid protein that is made in bone and binds avidly to the hydroxyapatite mineral. OCN is produced and secreted almost exclusively by osteoblasts, terminally differentiated cells responsible for the synthesis and mineralization of the bone matrix during the development of the skeleton and its periodic regeneration throughout life. Osteoblasts are short-lived cells originating from mesenchymal progenitors that are replaced depending on the demand for bone formation in a particular location and time. OCN secreted by osteoblasts contains three γ-carboxyglutamic acid residues that are removed by the acidic pH created during the resorption of bone by osteoclasts.

According to ideas originated and propagated by Gerard Karsenty and co-workers, osteoblasts comprise an endocrine organ and decarboxylated OCN is a hormone. The circulating levels of this “new hormone” are, therefore, dependent on the rate of bone turnover, a.k.a. remodeling. Specifically, Karsenty and collaborators have claimed in a series of high profile publications over a period of several years that OCN acts on multiple organs and tissues including bone, pancreas, liver, adipose, muscles, testicles, and the brain to regulates functions ranging from bone mass accumulation, to body weight, adipocity, glucose metabolism, energy utilization, male fertility, mentation, and behavior. Notably, all these claims have been based on the results of studies of a single mouse model with genetic deletion of OCN, generated in the Karsenty laboratory over 23 years ago, but never made available to other investigators seeking to reproduce the results.

However, unlike hormones produced and released by dedicated cells in response to external stimuli, the number of osteoblasts and thereby the circulating levels of osteocalcin inexorably change throughout life as a result of physiologic or pathologic changes of bone itself that can be acute or chronic, systemic or localized, and reversible or irreversible. Examples are skeletal development, growth, adaptation of the skeleton to mechanical forces, fracture healing, changing calcium needs, stress, menstrual cycle, pregnancy, lactation, menopause, aging, hyper- or hypo-parathyroidism, hyperthyroidism, hypercortisolemia, Paget’s disease, bone tumors, etc. Furthermore, medications – approved after extensive trials with thousands of subjects and used by millions for the treatment of osteoporosis – decrease or increase serum osteocalcin levels without any effect on glucose homeostasis, testosterone production, muscles, or behavior. The glaring shortcomings and incongruence of the Karsenty ideas with physiology, pathophysiology, clinical medicine, and pharmacology notwithstanding, the concerns with the hypothesis that bone and osteocalcin regulate many other tissues have progressively intensified over the years by the fact that other groups have failed to reproduce the observations of Karsenty and colleagues in different mice or in a rat model with OCN deletion. Still major journal review articles and prestigious teaching textbooks have continued until now to publish the Karsenty claims as proven biologic facts.

In the article by Moriishi and co-workers, the authors have generated their own OCN deficient mouse, using gene targeting in embryonic stem cells and microinjection into C57BL/6 blastocysts. Using this model they investigated the role of OCN on bone formation and mineralization, as well as glucose metabolism, testosterone production and muscle mass. They report that, in contrast to the results reported by Karsenty and colleagues, OCN plays no role in bone formation (or resorption) and the accumulation on bone mass in the estrogen sufficient or the estrogen deficient state. Instead, OCN is indispensable for the alignment of biological apatite crystallites parallel to collagen fibers. Loss of OCN function had no effect on collagen orientation, which remained normal. Bone strength, however, was decreased in the OCN deficient mice indicating that alignment of crystallites with collagen fibers is one of the elusive determinants of bone quality that together with bone mass determines the ability of bone to resist fractures. Additionally, Moriishi and colleagues show that OCN plays no role in the exercise-induced bone formation, glucose metabolism, the improvement of glucose metabolism caused by exercise, testosterone synthesis and spermatogenesis, or muscle mass.

This is an exceptionally thorough and careful investigation of the effects of OCN on bone as well as the earlier suggestions the OCN acts as hormone to regulate glucose metabolism, fertility, and muscle mass. The methods of analysis are flawless, the results (with the exception of the OVX experiment) are unambiguous, and the conclusions are fully justified and convincing. The writing of the manuscript, however, is suboptimal. There are unnecessary repetitions, stylistic issues and problems with the grammar that need to be addressed so as to improve the readability of this, otherwise, very impressive work.

Specific comments

1. Legend of Figure 1 H and I: In this and the legend of several other figures the X symbol above or below the box plots, presumably outliers, has not been defined.

2. Supplemental figure 2D: Cortical thickness and remodeling markers were not affected by OVX in the wild type control mice of this experiment (OVX at 5 weeks and sacrifice at 11 weeks). These results are in contrast to the well documented effect of OVX in adult mice that have already reached peak bone mass (4-5 months of age). The authors need to address this discrepancy and acknowledge the likelihood that during the experimental estrogen deficiency in their mice the femurs continued to grow, thereby confounding a meaningful interpretation.

Grammatical and stylistic suggestions:

1. The title should be shortened as follows: Osteocalcin is necessary for the alignment of apatite crystallites, but not glucose metabolism, testosterone synthesis, or muscle mass.

2. Osteocalcin should be abbreviated throughout the text as OCN. The use of articles, like a, as in “a Real-time PCR analysis” or “ A bone histomorphometric analysis” is unnecessary.

3. Abstract: I took the liberty to suggest the following small edits for the sake of clarity and brevity: Osteocalcin (OCN) which is specifically produced by osteoblasts and is the most abundant non-collagenous protein in bone. It has been shown suggested previously that OCN shown to inhibits bone formation. It also and functions as a hormone that to regulates insulin secretion in the pancreas, testosterone synthesis in the testes, and muscle mass. We have generated a strain of OCN deficient mice using gene targeting in embryonic stem cells and microinjection into C57BL/6 blastocysts. However, an analysis of Ocn–/– mice revealed Loss of OCN function had no effect on the regulation of bone quantity, glucose metabolism, testosterone synthesis, or muscle mass in these mice. OCN deletion, however, disrupted the alignment of biological apatite crystallites parallel to collagen fibers and caused loss of strength, indicating that alignment of crystallites with collagen fibers is one of the determinants of bone quality. that together with bone mass determines the ability of bone to resist fractures. Although the orientation degree of collagen fibrils and size of biological apatite (BAp) crystallites in the c-axis were normal in Ocn–/– bone, the crystallographic orientation of the BAp c-axis, which is normally parallel to collagen fibrils, was severely disrupted, resulting in reduced bone strength. These results demonstrate that osteocalcin is required for maintaining bone quality and strength by adjusting the alignment of BAp crystallites parallel to collagen fibrils.

4. Author Summary: Here again, I’ve taken the liberty to suggest the following small edits. The strength of bone is dependent on both bone quantity and quality. Osteocalcin (OCN) is the most abundant non-collagenous protein in bone. It has been suggested before that OCN inhibits bone formation, and serum osteocalcin has been shown to and works as a hormone that to regulates glucose metabolism, testosterone synthesis, and muscle mass. We have generated osteocalcin-deficient mice, and found that osteocalcin is not involved in the regulation of bone quantity. The major factors that influence bone quality are the alignment of collagen fibers and mineralization. Collagen fibers align parallelly to the longitudinal axis of bone and mineralization occurs by the growth of apatite crystallites parallel to the collagen. We showed that In our osteocalcin-deficient mice collagen fibers aligned normally but apatite crystallites aligned randomly against collagen, resulting in reduced bone strength. Further, we found that all of the data in found that glucose metabolism, testosterone synthesis, and muscle mass are normal in osteocalcin-deficient these mice. We demonstrated conclude from this work that OCN aligns apatite crystallites parallelly to collagen fibers and thereby it maintains bone quality and strength. but that sSerum osteocalcin, however, does not work as a hormone and does not affect regulates glucose metabolism, testosterone synthesis, and muscle mass.

5. Introduction, pages 3-7: This entire section is long, verbose, and unnecessarily detailed. It should be shortened and simplified, for example by using review articles to summarize the earlier findings/claims about OCN.

6. Introduction, 1st paragraph: Fracture risk depends on many factors besides the quantity and quality of bone, for example falls. Gla should be defined the first time it is mentioned on line 7, but in any case the use of the terms Gla and Glu is confusing and unhelpful. Carboxylated and de-carboxylated or un-corboxylated OCN are much simpler and easier terms for the reader to follow. Carboxylated OCN has (instead of exhibits) high affinity for (instead of to) Ca.

7. Introduction, last paragraph: this is an unnecessary repetition of the abstract and summary.

8. Page 17, last two lines to end of this paragraph on page 18: the way this paragraph is written it implies that the earlier claims are factual. This is confusing. Moreover, several of the thoughts discussed here are repetition of earlier sections.

Reviewer #2: The authors constructed mice knocked out for the two mouse osteocalcin genes and the DNA between the osteocalcin genes in a manner that appears similar to the strategy initially used by Ducy et al. they show that bone mass, bone histomorphometry, serum markers of bone formation and resorption are similar between WT and KO mice. Using EM analysis of antibody treated sections they show that osteocalcin is located in both the intrafibrillar and interfibrillar regions of collagen in WT mice and not detected in the KO. Raman microspectroscopy also revealed no differences in the collagen and mineral chemical indices. However, the c-axis orientation degree of bone apatite was disrupted in the KO mice compared to WT; this correlated with a lower Yong’s modulus in single and multiple regression analysis, after nanoindentation testing. Three point bending analysis was normal in the KO mice. Glucose and insulin tolerance tests were normal in the KO mice, as was their weight and blood glucose. In normal mice exercise improves glucose metabolism and bone formation. Male KO mice had normal sperm and testosterone levels and were fertile. Indices of muscle mass were normal in the KO. Thus, the authors fail to confirm several of the previous findings from the Karsenty group’s studies of similar mice.

1. The wording of the conclusion at the end of page 18 is too sweeping. The authors show that removal of the osteocalcin gene shows that osteocalcin is not needed for normal bone formation, glucose metabolism, or male testosterone production. This is an important and novel series of observations. But, they did not test and cannot comment on the ability of osteocalcin to regulate bone formation, glucose metabolism, testosterone synthesis or the maintenance of muscle mass. It is always possible that other redundant hormones and paracrine factors can compensate for the absence of osteocalcin in the KO mice. So, the authors need to be more constrained in their summary claim.

2. Figure 9G. It is not clear what statistical analysis led to the conclusion that the blue dots, which appear to be higher than the red dots, are not statistically different.

3. Figure 10. There’s no mention I can find of what the asterixes in this figure signify.

4. It might be useful for the authors to comment on why the results from their studies differ from those of the Karsenty group. For example, are the knockouts exactly the same? I can’t tell from a simple comparison of the papers. If some sequence adjacent to the coding regions for osteocalcin have effects on another gene, for example, then perhaps that could explain differences. Or maybe different versions of C57B mice could make a difference.

Reviewer #3: This is a Herculean manuscript that describes the KO phenotype of a Blglap/Bglap2 KO mouse. The Ocn-/- mouse phenotype is of great interest due to a series of high profile publications on a previous Ocn-/- mouse line, which report a number of key roles for Ocn in regulating fertility and metabolism.

The authors perform a number of studies in great detail, including both male and female mice of different ages in studies of bone structure, glucose metabolism, and fertility. The authors comprehensively demonstrate a lack of phenotype for their Ocn-/- mouse in the above endpoints – bar an effect on alignment of apatite crystals and sequelae.

Overall, I comment the authors and have few, if any, comments on the studies performed. The results seem very clear. Find below some considerations and minor questions:

1. In general the statistical analyses are not well described; please clarify the nature of the ANOVA performed (in some cases it could be one-, two-, or three-way ANOVA), and give the overall P value for the genotype effect (which, looking at the graphs, would be absent for most of the figures). Also, I presume that the “x” symbols on the graphs are either outliers, or limits of the data range beyond which would be outliers? Please clarify – and if outliers, describe the method used to exclude.

2. Fig 4 – please clarify how these images were selected for the figure (as relatively few Ocn immunoreactive dots are visible in field).

3. Please comment on the potential role (or lack thereof) of Bglap3 – and any differences, or not, with respect to previous Ocn-/- mice lines?

4. Fig 7 – correlation of YM and Bap c-axis orientation – whilst there is a significant correltion, the points are rather clustered at each end of the regression – so I would interpret with caution.

5. Given the controversy in the field, it might be useful if the authors make a written statement that they will share their new mouse line with other researchers?

**Have all data underlying the figures and results presented in the manuscript been provided?**

Reviewer #1: Yes

Reviewer #2: Yes

Reviewer #3: Yes

PLOS authors have the option to publish the peer review history of their article (what does this mean?). If published, this will include your full peer review and any attached files.

Reviewer #1: Yes: Stavros C. Manolagas, MD, PhD

Reviewer #2: No

Reviewer #3: No

---

## [Decision Letter · Decision Letter 1]

21 Nov 2019

Dear Dr Komori,

Thank you very much for submitting your Research Article entitled 'Osteocalcin is necessary for the alignment of apatite crystallites, but not glucose metabolism, testosterone synthesis, or muscle mass' to PLOS Genetics. Your manuscript was fully evaluated at the editorial level and by independent peer reviewers.

It is requested that you address the two minor issues raised by reviewer 3.  In light of this reviewer’s comments it is important for the authors to consider the appropriateness of the particular statistical test used and to also provide the overall p value for the ANOVA.  The authors should also include a comment on the objective selection of mineralized areas displayed in Fig 4.  Reviewer 1 has added comments for consideration for modification of the text in several places.  The authors should consider whether some of these suggestions may improve the clarify of the manuscript.

We therefore ask you to modify the manuscript according to the review recommendations before we can consider your manuscript for acceptance. Your revisions should address the specific points made by each reviewer.

[LINK]

Yours sincerely,

John F Bateman

Associate Editor

PLOS Genetics

Gregory Barsh

Editor-in-Chief

PLOS Genetics

Reviewer's Responses to Questions

**Comments to the Authors:**

Reviewer #1: The revisions of the manuscript have improved its clarity, but few areas need some additional editing, especially in the authors’ summary and abstract. I have taken again the liberty to suggest some editorial and grammatical changes. To facilitate the authors, the manuscript PDF has been edited and attached to the review.

Reviewer #2: The authors have addressed my concerns.

Reviewer #3: The authors have addressed the main points from the initial review, however, a couple of clarifications would be useful:

Re the ANOVA used, could the authors please clarify why only one-way ANOVA analysis has been used? To take an example, in Figs 11B and 11C, there are two variables (age and genotype), which implies a two-way ANOVA should be preferred? In respect to the request for P values, the request was NOT for the individual P values for the post-hoc analyses (I agree, this would make the graphs very crowded). The suggestion was that the overall P value for the ANOVA (e.g. effect of genotype) be included in the text or figure legend. Not compulsory, but it provides a level of granularity that compliments the graphical data.

Secondly, re the selection of images (Fig 4). The authors state that "Completely mineralized regions with some distance from osteoid were selected". Just for transparency, it would be useful if the authors can make. statement to the effect that the areas were selected based on an objective criterion, and are not subject to any bias. (FWIW, I am not suggesting that they are - I would just like a statement that confirms their objective selection).

**Have all data underlying the figures and results presented in the manuscript been provided?**

Reviewer #1: Yes

Reviewer #2: Yes

Reviewer #3: Yes

PLOS authors have the option to publish the peer review history of their article (what does this mean?). If published, this will include your full peer review and any attached files.

Reviewer #1: Yes: Stavros Manolagas, MD, PhD

Reviewer #2: No

Reviewer #3: No

---

## [Editor Report · Decision Letter 2]

29 Dec 2019

Dear Dr Komori,

We are pleased to inform you that your manuscript entitled "Osteocalcin is necessary for the alignment of apatite crystallites, but not glucose metabolism, testosterone synthesis, or muscle mass" has been editorially accepted for publication in PLOS Genetics. Congratulations!

Yours sincerely,

John F Bateman

Associate Editor

PLOS Genetics

Gregory Barsh

Editor-in-Chief

PLOS Genetics

Comments from the reviewers (if applicable):

**Data Deposition**

http://datadryad.org/submit?journalID=pgenetics&manu=PGENETICS-D-19-01071R2

**Press Queries**

---

## [Editor Report · Acceptance letter]

28 Apr 2020

PGENETICS-D-19-01071R2 

Osteocalcin is necessary for the alignment of apatite crystallites, but not glucose metabolism, testosterone synthesis, or muscle mass 

Dear Dr Komori, 

We are pleased to inform you that your manuscript entitled "Osteocalcin is necessary for the alignment of apatite crystallites, but not glucose metabolism, testosterone synthesis, or muscle mass" has been formally accepted for publication in PLOS Genetics! Your manuscript is now with our production department and you will be notified of the publication date in due course.

With kind regards,

Kaitlin Butler

PLOS Genetics

On behalf of:
